# Dissecting Multimodal In-Context Learning:
# Modality Asymmetries and Circuit Dynamics in modern Transformers

Yiran Huang [1 2 3]   Karsten Roth [4]   Quentin Bouniot [1 2 3 5]   Wenjia Xu [6]   Zeynep Akata [1 2 3]

## Abstract

Transformer-based multimodal large language models often exhibit in-context learning (ICL) abilities. Motivated by this phenomenon, we ask: **how do transformers learn to associate information across modalities from in-context examples?** We investigate this question through controlled experiments on small transformers trained on synthetic classification tasks, enabling precise manipulation of *data statistics* and *model architecture*. We begin by revisiting core principles of unimodal ICL in *modern* transformers. While several prior findings replicate, we additionally find that scaling up favours in-weight memorisation over ICL, and that Rotary Position Embeddings (RoPE) increases the data complexity threshold for ICL. Extending to the multimodal setting reveals a fundamental learning asymmetry: when pretrained on high-diversity data from a primary modality, surprisingly low data complexity in the secondary modality suffices for multimodal ICL to emerge. Mechanistic analysis shows that both settings rely on an induction-style mechanism that copies labels from matching in-context exemplars; multimodal training refines and extends these circuits across modalities. Code is available at: https://github.com/YiranHuangIrene/multimodal-icl

## 1. Introduction

In-context learning (ICL), the ability to perform new tasks from demonstrations without parameter updates (Radford et al., 2019; Brown et al., 2020), has emerged as a core capability of transformer-based large language models (LLMs). Understanding the origins and mechanisms of this capability has become a central focus of transformer research.

For unimodal ICL, progress has been made on both fronts. On the training side, studies have identified statistical properties of pretraining data, such as burstiness, high class diversity, and skew, that promote ICL over in-weight learning (IWL) (Chan et al., 2022; Singh et al., 2023; Chan et al., 2024; Zucchet et al., 2025). Fundamentally, these mechanisms compete to minimize loss: while IWL relies on memorizing static input-label mappings in parameters, ICL emerges when high data diversity makes such memorization inefficient, forcing the model to rely on context. On the mechanistic side, researchers have traced ICL to specialized *induction circuits*: attention patterns that retrieve contextually relevant examples and copy their associated labels to the query (Olsson et al., 2022; Crosbie & Shutova, 2025; Reddy, 2024). However, these mechanistic insights derive primarily from simplified transformers that lack key architectural components of modern LLMs, such as RoPE.

Recent advances in multimodal large language models (MLLMs) (Alayrac et al., 2022; Abdin et al., 2024; Hui et al., 2024) have extended ICL to interleaved image-text demonstrations, enabling cross-modal reasoning from contextual examples alone. This observation raises a fundamental question about transformer learning:

*What enables transformers to perform multimodal ICL, both in terms of training data statistics and mechanisms?*

To answer this question, this paper provides a systematic, reverse-engineering account of multimodal ICL in modern transformers. We investigate how statistical principles governing unimodal ICL in simplified transformers (Reddy, 2024) transfer to modern LLM-style architectures, how they extend to multimodal settings, what underlying attention circuits implement multimodal ICL, and how they relate to their unimodal counterparts. To isolate causal factors, we train small but architecturally realistic two-layer transformer models on controllable synthetic classification tasks. This controlled environment allows us to systematically vary data statistics and architectural components while tracking the formation of attention patterns throughout training—an anal-

---

[1]Technical University of Munich [2]Helmholtz Munich [3]Munich Center for Machine Learning [4]DeepMind [5]LTCI, Télécom Paris, Institut Polytechnique de Paris [6]Beijing University of Posts and Telecommunications. Correspondence to: Yiran Huang <yiran.huang@tum.de>.

*Proceedings of the 43$^{rd}$ International Conference on Machine Learning*, Seoul, South Korea. PMLR 306, 2026. Copyright 2026 by the author(s).

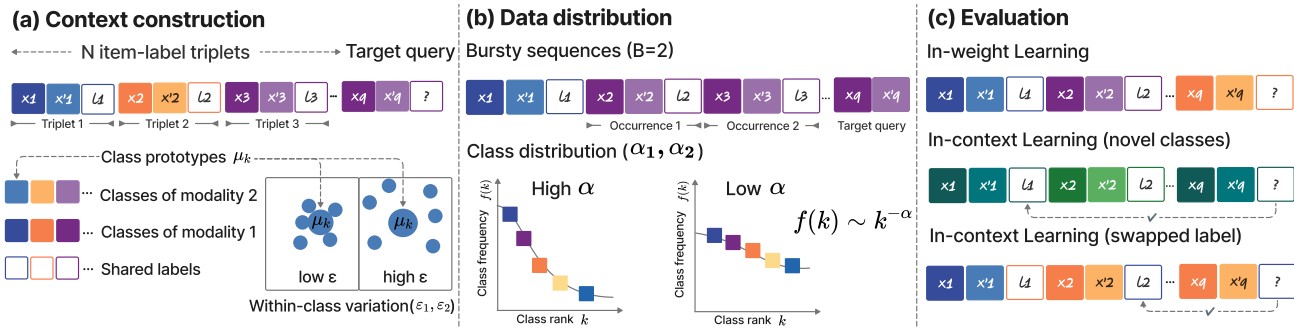

*Figure 1.* Preliminaries in the multimodal setting. **(a)** The context consists of $N$ triplets followed by the target query. The paired examples $(x_i, x'_i)$ from two modalities, with a shared label $l_i$, are generated from Gaussian Mixture Models (GMMs) by controlling within-class variation $\varepsilon_1$ and $\varepsilon_2$. **(b)** The distributional properties for the synthetic data. The burstiness $B$ determines how often the class occurs in the context. Class frequencies follow a Zipfian distribution with exponents $\alpha_1$ and $\alpha_2$. **(c)** Evaluation distinguishes between IWL, where target queries belong to class seen during training while not in the context during evaluation, and ICL, where target queries are novel but in the context. A swapped-label condition further isolates ICL by permuting the labels.

ysis that would be impossible with real-world pretraining corpora, where multiple variables are confounded.

Our investigation yields the following key findings: (1) While the statistical drivers of unimodal ICL in simplified transformers transfer to modern architectures, RoPE raises the data complexity threshold for ICL. Furthermore, scaling up unimodal models favors in-weight memorization over ICL at fixed data complexities (Sec. 3). (2) Multimodal ICL exhibits a strong learning asymmetry. When the model is pretrained on high-diversity data from a primary modality, surprisingly low data complexity in the secondary modality suffices for multimodal ICL to emerge (Sec. 4.1). Consequently, and unlike the unimodal case, scaling consistently improves multimodal ICL. The added capacity is utilized to map the secondary modality to the pre-existing circuit rather than for memorization (Sec. 4.2). (3) Multimodal ICL performance is bottlenecked by the modality alignment; a pretrained encoder is important for achieving this alignment (Sec. 4.3). (4) Beyond synthetic data, we validate our distributional findings and the importance of an encoder on Omniglot (Lake et al., 2019) and Mini-ImageNet (Russakovsky et al., 2015). (5) Mechanistically, induction circuit strength tracks ICL accuracy in both unimodal and multimodal settings. Multimodal training primarily refines the induction head responsible for label matching (Sec. 4.4). (6) These mechanistic claims transfer to production scale: in Qwen2.5-VL-3B (Team, 2025), the previous-token and induction heads identified in our testbed largely overlap with the LLM backbone's, causally drive ICL accuracy under head knockout, and refine during fine-tuning exactly as in the synthetic setting (Sec. 5).

In summary, our contributions are: (1) the first mechanistic account of multimodal ICL in transformers; (2) a controlled testbed for investigating how architectural choices and data statistics shape ICL across modalities; (3) the discovery of learning asymmetry between modalities and characteriza-

tion of the underlying circuit dynamics; and (4) a validation on the MLLM demonstrating that the identified circuits are inherited, causally responsible, and dynamically refined under fine-tuning.

## 2. Preliminaries

**Task description.** Let $\mathcal{X}_1$ and $\mathcal{X}_2$ denote the input spaces of two modalities and let $\mathcal{L} = \{L_1, \ldots, L_n\}$ be the shared label set. We feed the model with a context consisting of $N$ labelled exemplars followed by an unlabelled query. In the unimodal setting, we follow Reddy (2024), with context comprising $N$ item–label pairs from a single modality: $x_1, \ell_1, x_2, \ell_2, \ldots, x_N, \ell_N, x_q$. Each $x_i \in \mathcal{X}_1$ is an example whose ground-truth label is $\ell_i \in \mathcal{L}$. The model must predict $\ell_q$ for the query item $x_q$ (see App. A.1). In the multimodal setting, we extend the task by presenting paired exemplars from two modalities: $x_1, x'_1, \ell_1, x_2, x'_2, \ell_2, \ldots, x_N, x'_N, \ell_N, x_q, x'q$. Here $x_i \in \mathcal{X}_1$ and $x'_i \in \mathcal{X}_2$ correspond to the same label $\ell_i$ (see Fig. 1a). Importantly, at least one exemplar (unimodal) or exemplar pair (multimodal) in the context shares the query's class, ensuring that ICL is in principle possible.

**Synthetic data generation.** We consider two modalities and generate data for both modalities from Gaussian Mixture Models (GMMs), allowing for precise control over data properties. For modality $m \in \{1, 2\}$, we sample from a GMM with $K_m$ classes in $\mathbb{R}^{D_m}$. Class prototypes are $\mu_k \sim \mathcal{N}(0, I_{D_m}/D_m)$, and class instances are generated by:

$$x_i = \frac{\mu_k + \varepsilon_m \eta}{\sqrt{1 + \varepsilon_m^2}}, \quad \text{where} \quad \eta \sim \mathcal{N}(0, I_{D_m}/D_m). \quad (1)$$

The parameter $\varepsilon_m$ sets the within-class variability (Figure 1a), with rescaling factor ensuring that $\|x\| \approx 1$. Each modality has $K_m$ classes mapped many-to-one into the shared label set $\mathcal{L}$. This models fine-grained subclasses

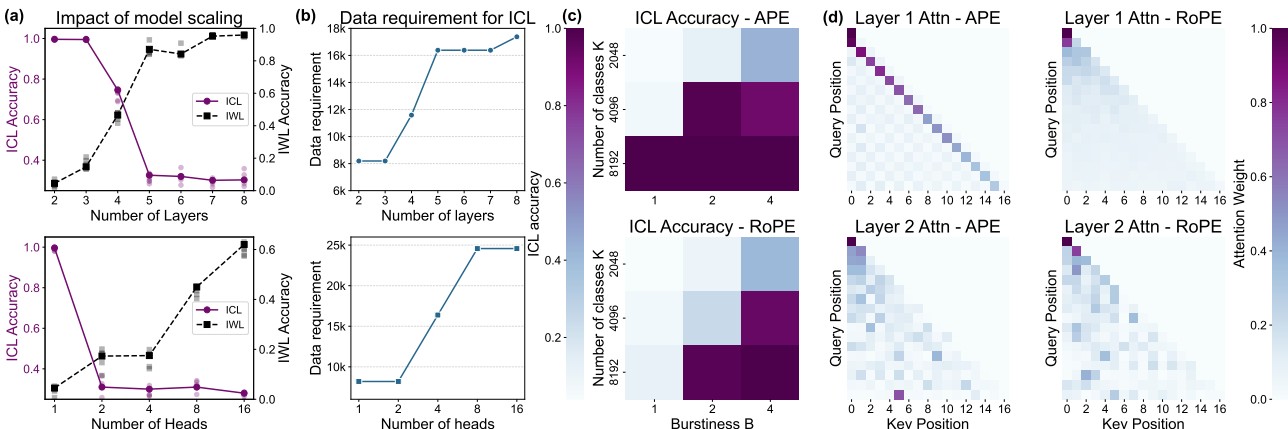

*Figure 2.* (a) Impact of increasing model layers and attention heads on ICL-IWL tradeoff. Scaling up favours IWL over ICL for five seeds. (b) The data complexity (measured by $K \cdot \sqrt{B}$) required for models of different sizes to achieve the same ICL accuracy ($> 0.95$). A larger model needs more complex data for a strong ICL capability. (c) Across data regimes, RoPE yields lower ICL accuracy than absolute positional encodings (APE). (d) Attention maps for an example with the correct label at position 5: absolute PE shows clear previous-token and induction heads; with RoPE these patterns are diminished. All runs use unimodal training with $K = 8192, B = 1, \alpha = 0$ except when that parameter is varied.

collapsing to a common output vocabulary. Each label is also associated with a prototype vector sampled from $\mathcal{N} \sim (0, I_D/D)$. In the multimodal setting, the two generators are dependent. We set the label space $\mathcal{L}_2 \subset \mathcal{L}_1$ to reflect common practice where the decoder generates in the primary modality's vocabulary (e.g., text), so the secondary modality aligns to that vocabulary rather than expanding it. We use $N = 8, L_1 = 32, L_2 = 16, D_1 = 64, D_2 = 32, \epsilon_1 = \epsilon_2 = 0.1$ unless otherwise specified.

**Parameterizing the data distribution.** Beyond the intrinsic GMM parameters ($D, K, \varepsilon$), we also control sequence-level statistics. The *burstiness* $B$ (Figure 1b) creates contexts with $N/B$ classes, each appearing $B$ times. This models the multi-shot demonstrations common in real prompts. *Zipfian skew* $\alpha$ (Piantadosi, 2014) parameterizes long-tailed class frequencies $f(k) \sim k^{-\alpha}$ (Figure 1b), determining the balance between frequent and rare concepts.

**Training Protocol.** We train all models using the SGD optimizer with a batch size of 128. We utilize a learning rate of $1 \times 10^{-3}$ and a weight decay of $1 \times 10^{-6}$. To ensure robust comparisons, all models are trained until convergence.

**Evaluation metrics.** We explicitly separate two learning mechanisms to avoid misattributing memorization (IWL) to contextual reasoning (ICL). IWL assesses performance on test sequences sampled i.i.d. from the training distribution, measuring knowledge stored in parameters, while ICL assesses performance on sequences with novel classes, forcing reliance on contextual examples rather than memorized associations. As an additional ICL metric, we evaluate the model on sequences in which the context labels are swapped relative to training, invalidating the memorized mapping. We illustrate the evaluation in Figure 1c and App. A.1. To

ensure robustness, all reported experiments are averaged over 5 random seeds. For heatmaps, standard deviations are generally $< 0.03$ unless otherwise noted in the appendix.

# 3. Establishing Architectural Premises: ICL in Modern Transformers

While previous work (Reddy, 2024) has revealed foundational properties of ICL in attention-only transformers, such as dependency on specific data distribution, it remains unclear how these principles operate within modern, LLM-style configurations. Here, we construct a controlled setting to isolate the data-centric and architecture-centric factors that drive ICL, providing premises for the multimodal study. Concretely, we move beyond the attention-only setup and adopt a two-layer transformer decoder architecture incorporating RMSNorm (Zhang & Sennrich, 2019), SiLU (Elfwing et al., 2018), and RoPE (Su et al., 2022), which are components characteristic of contemporary LLMs (Touvron et al., 2023). We systematically vary the statistical properties of the training data and model size, enabling clean attribution of how each factor shapes the emergence of ICL.

## 3.1. Data statistical principles hold within modern architectures.

All key findings revealed in Reddy (2024); Chan et al. (2022) are reproduced: increasing the number of classes $K$, burstiness $B$, or within-class variation $\varepsilon$ promotes ICL, while increasing the Zipf exponent $\alpha$ shifts performance toward IWL, with an optimal balance achieved when $\alpha = 1$ (see App. A.2.1). We also recover the ICL–IWL trade-off: settings that strengthen ICL correspondingly diminish IWL,

and vice versa. Together, these findings confirm that the core data statistical drivers are architecture-agnostic.

## 3.2. Model scaling raises the ICL threshold.

We systematically scale up model depth or the number of attention heads and train the model until convergence, using data with the same statistical cues, i.e., same data complexity. We reveal a fundamental tension between model capacity and ICL utilization. Given the same data complexity, larger models tend to favor IWL solutions over ICL ones (Figure 2a). They require substantially stronger statistical cues in training data to achieve the same ICL accuracy (Figure 2b). This requirement grows faster with the number of heads than with the number of layers. We attribute this to how multi-head attention distributes capacity: more heads allow information to be partitioned into specialized subspaces, enabling the item–label memorization. This creates a low-loss shortcut that competes with the ICL solution. Adding layers increases depth but does not create as many independent storage slots for memorization, so the bias toward in-weight learning is weaker. This interpretation aligns with prior observations that induction-style behavior typically emerges in a subset of heads while other heads specialize in memorization (Elhage et al., 2021; Singh et al., 2024). Importantly, *scaling does not eliminate ICL capacity*; it raises the threshold of statistics cues needed for the ICL solution to outcompete memorization.

## 3.3. RoPE raises the data complexity threshold for ICL.

Switching from absolute positional encodings (APE) to RoPE causes a marked drop in ICL accuracy (Figure 2c). Although RoPE is widely adopted for its strong length generalization, in our setting, this benefit coincides with a consistent degradation of ICL. Attention visualizations show that the model struggles to form strong previous-token heads with RoPE, and the induction head is also less clear (Figure 2d), which we will investigate in Sec. 4.4. We hypothesize this is because RoPE's multiplicative rotational structure lacks the discrete, offset-based cues of absolute encodings, making it harder for the model to learn the simple token-copying operations crucial for ICL. For completeness, we also evaluate ALiBi (Press et al., 2021) and Hybrid PE. We provide detailed results including attention circuit analysis with different positional encodings in App. A.2.2, which show that the design of relative positional encodings provides a weaker inductive bias for the simple, offset-based induction circuits. This effect is most pronounced in low-data-complexity regimes, and *sufficiently high data complexity can compensate for this weaker bias.*

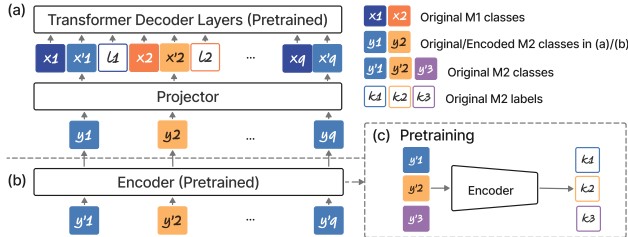

*Figure 3.* Multimodal setup. (a) Projector-only setup: an MLP projector aligns M2 features to the M1 embedding space. (b) Encoder-augmented setup: a pretrained M2 encoder is stacked before the projector and decoder. (c) Encoder pretraining: the M2 encoder is pretrained on M2-specific classes/labels.

> **Takeaway**: Data distributional effects on ICL persist in modern LLM-style transformers. Scaling shifts learning toward in-weight memorization. RoPE further suppresses ICL by disrupting induction circuits.

## 4. Multimodal In-context Learning

Building on our unimodal ICL baseline, we now investigate the multimodal setting to understand how ICL principles transfer across modalities. We employ a two-stage training process (Figure 3). First, we pretrain a transformer decoder, same as the unimodal setting, on a single primary modality M1. Next, we introduce a secondary modality M2 by adding a simple MLP projector to map M2 features into M1's embedding space. We then jointly train it with the pretrained decoder. As an optional extension, we insert a pretrained M2 encoder before the projector to probe representation quality. We use $K_1 = 8192, K_2 = 256, B = 4, \epsilon_1 = \epsilon_2 = 0.1, \alpha_1 = \alpha_2 = 0$ unless otherwise specified.

### 4.1. Data Asymmetries Reveal a Primary and Secondary Role for Modalities.

We examine how the data distributions of two modalities affect ICL. Firstly, the results (Figure 4a) show that increasing the class number and burstiness of M2 increases ICL, similarly to the unimodal setting. Moreover, our experiments reveal a fundamental asymmetry: as shown in Figure 4a, after pretraining the decoder on a high-diversity M1 ($K_1 = 8192$), M2 requires surprisingly little class diversity to achieve comparable ICL; a relatively small $K_2 = 256$ is sufficient. This suggests that M1's role is to install the core ICL capability, while M2's primary role is simply to provide a distinguishable signal that the projector can map onto the decoder's pre-existing feature space. Further experiments support this view. While within-class variation $\varepsilon$ in both modalities boosts ICL, increasing it for M2 has a much stronger positive effect (Figure 4c). Since M1 is already well-represented from pretraining, higher $\varepsilon_2$ is more critical as it forces the model to learn *a robust and generalizable mapping* for the new modality. Finally, performance

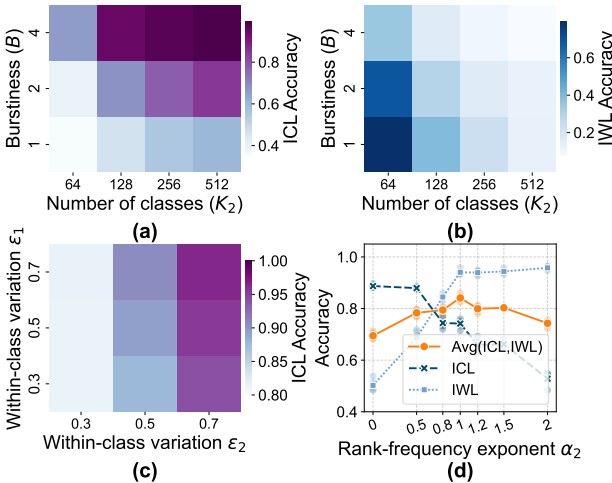

**Figure 4.** Multimodal asymmetry. (a) Fixing $K_1 = 8192$, the impact of $K_2$ and $B$ on ICL performance. When the decoder is pretrained on M1, a significantly lower data complexity for M2 is needed to achieve good ICL. (b) Consistently, IWL decreases with $K_2$ and $B$. (c) Raising $\varepsilon_2$ benefits ICL markedly more than raising $\varepsilon_1$. (d) Fixing $\alpha_1=1$, the ICL–IWL balance is best when $\alpha_2 \approx 1$ for five seeds.

is optimized when class-frequency skews ($\alpha$) match across modalities (Figure 4d): pretraining M1 with $\alpha_1 \approx 1$, the Zipfian skew characteristic of natural-language corpora used to pretrain decoders (Piantadosi, 2014), yields the best multimodal balance when M2 also has $\alpha_2 \approx 1$.

**Structural origins of asymmetry.** We provide further evidence that the observed primary/secondary hierarchy is a result of the training curriculum. We conducted early-fusion joint training experiments where the model is trained on both modalities from scratch, while strictly maintaining the same interleaved sequence structure $(x_i, x_i', l_i)$. We find that removing the pretraining stage reverses the learning asymmetry: the model becomes significantly more sensitive to the data complexity of M2 rather than M1. Detailed results are provided in App. A.3.7. This result confirms that the dominance of M1 in our main analysis is driven by the pretraining phase, which installs the initial ICL circuit. In the absence of this pretraining bias, the model naturally anchors the induction circuit to M2 due to its architectural advantage of being positionally adjacent to the label.

## 4.2. The Decoder's Architecture Has Contrasting Effects on Multimodal ICL

**Scaling up the decoder favors multimodal ICL.** We find that larger models can achieve strong ICL (accuracy $>$ 95%) with significantly lower data complexity requirements (Figure 5). We attribute this to the increased representational bandwidth available in larger models. Since the decoder is initialized with ICL capabilities from the primary modality M1, this added capacity allows it to successfully integrate

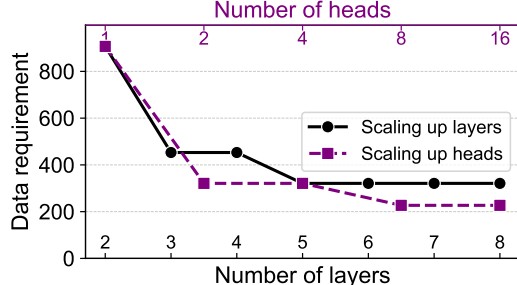

**Figure 5.** Data requirements (measured by $K_2 \cdot \sqrt{B}$) needed for larger multimodal models to elicit strong ICL. Deeper or wider decoders achieve the same accuracy with lower data requirements.

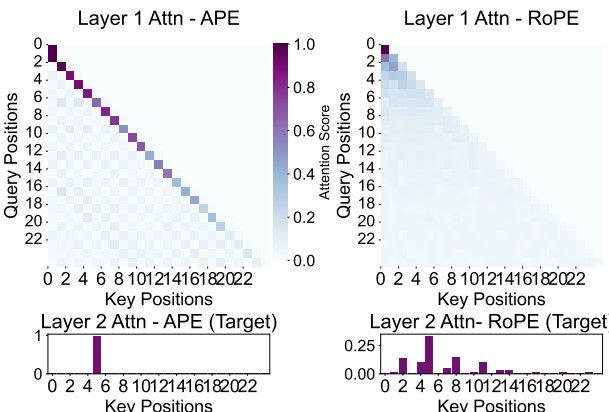

**Figure 6.** The induction mechanism is preserved in the multimodal setting. Both APE and RoPE exhibit characteristic previous-token copying (Layer 1) and induction spikes at the correct label index 5 (Layer 2). While RoPE blurs the pattern, the persistence of the peak confirms the mechanism transfers to the multimodal setting.

the secondary modality M2 via the projector, thereby enabling effective generalization to the multimodal task.

**RoPE continues to raise the data complexity threshold.** Consistent with our unimodal findings (Sec. 3.3), RoPE imposes a higher data requirement for ICL emergence in the multimodal setting (see detailed results in App. A.3.1). Interestingly, attention visualizations reveal a crucial mechanistic continuity. As shown in Figure 6, while APE yields sharp previous-token and induction patterns, RoPE exhibits the same mechanistic signature, assigning peak attention to the correct label token despite a more diffuse distribution. This confirms that the core ICL mechanism transfers across modalities, even if RoPE weakens its signal clarity. We provide a detailed quantitative analysis of these circuit dynamics in Sec. 4.4.

## 4.3. The Role of the Encoder: Cross-Modal Alignment and Feature Quality

While attention mechanisms drive ICL, they can only operate effectively on the representations they receive. Our analysis isolates two critical bottlenecks: cross-modal mis-

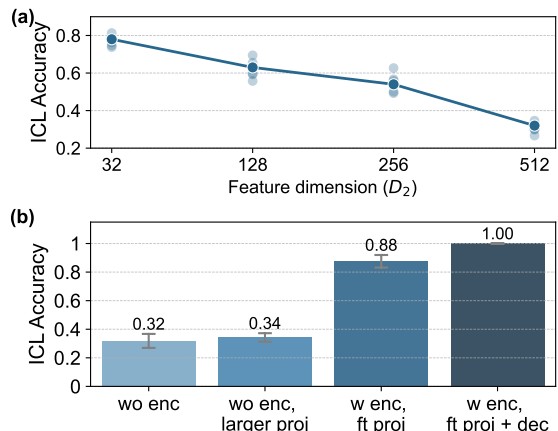

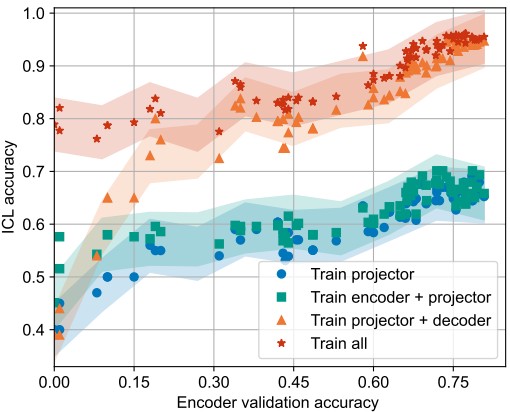

*Figure 7.* **Pretrained encoders enhance multimodal ICL.** (a) $D_1 = 64$, increasing M2 feature dimension ($D_2$) lowers ICL when using only a projector. (b) Fixing $D_2 = 512$, adding a pretrained encoder outperforms a parameter-matched larger projector.

*Figure 8.* **Encoder quality predicts downstream ICL.** Multimodal ICL on Omniglot increases with the encoder's validation accuracy; gains saturate only when the decoder is frozen, while joint training of the decoder continues to yield improvements.

alignment and feature discriminability. We demonstrate that the primary role of the encoder is to reshape M2 features to overcome the former, while the quality of the encoder determines the tractability of the latter. **The encoder bridges the misalignment gap.** Our initial experiments revealed a consistent drop in ICL accuracy as M2 feature dimensionality increased without any dedicated processing (Figure 7a). This suggests a "misalignment gap": the projector cannot extract sufficient structure from high-dimensional features to allow the decoder's induction heads to engage. To address this gap, we introduce a ViT encoder pretrained on M2 data (Figure 3c). We experiment with three training strategies: training (1) only the projector, (2) the projector & pretrained decoder, and (3) all the components. Results in Figure 7b indicate that the encoder significantly boosts ICL accuracy across all training regimes. This gain is not merely a function of added capacity; a model with a pretrained encoder consistently outperforms a parameter-matched baseline with a larger projector.

We attribute this improvement explicitly to better cross-modal alignment. Quantitative analysis of the feature space reveals that higher M2 feature dimensions ($D_2$) degrade the alignment between the two modalities: as $D_2$ increases from 32 to 512, Centered Kernel Alignment (CKA) drops from 0.16 to 0.07, and Euclidean distance ($L_2$) increases from 0.95 to 2.15. However, employing the encoder mitigates this misalignment at high dimensions ($D_2 = 512$), increasing CKA from 0.07 to 0.10 and reducing $L_2$ from 2.15 to 1.95. Full results are provided in Appendix A.3.2. This suggests the encoder acts as a bridge, compressing the M2 data into a manifold that aligns better with the decoder's latent space, thereby allowing the decoder to successfully apply its ICL capabilities to the new modality.

**Validation on real data reveals the importance of M2 representation quality.** While synthetic data offers fine-

grained control over distributional properties, it is significantly easier to classify compared to real-world inputs. To further investigate the role of the encoder under more realistic conditions, we extend experiments to Omniglot (Lake et al., 2019) and Mini-ImageNet (Russakovsky et al., 2015) images as M2. Concretely, we first show that all the data distributional findings transfer to real images (App. A.3.3), then train encoders with varying difficulty levels (detailed in App. A.3.4) and thus with different validation accuracies. Afterwards, we measure the impact of these encoders on downstream multimodal ICL.

As shown in Figure 8, encoder validation accuracy strongly predicts multimodal ICL performance. This indicates that the discriminative quality of the input features is critical. A stronger encoder groups similar inputs more tightly, providing a distinct signal that simplifies the projector's task. By reducing ambiguity in the input space, the encoder allows the projector to more accurately map M2 features to the target M1 embedding space. Furthermore, joint fine-tuning of the encoder, projector, and decoder yields the highest accuracy, indicating that maximizing ICL requires the co-adaptation of all the components.

> **Takeaway.** With a decoder initialized from unimodal pretraining, strong multimodal ICL is achievable without highly complex data from the second modality. Scaling up the decoder size improves ICL. Additionally, encoders enhance ICL by aligning compressed, discriminative features with the decoder.

### 4.4. Quantifying ICL Mechanisms

To understand how multimodal ICL emerges mechanistically, we move beyond accuracy curves and analyze the formation of specific attention circuits. Prior work on unimodal simple transformers identified a two-step ICL circuit:

an early-layer "previous-token head" copies information, and a later-layer "induction head" uses it to find matching examples and retrieve their labels (Olsson et al., 2022). However, these circuits can be difficult to detect when attention patterns are diffuse, particularly with RoPE. To quantify the circuit dynamics of ICL even when the patterns are not clearly visible, we employ a suite of progress measurements designed to track the formation of key attention patterns.

### 4.4.1. PROGRESS MEASUREMENTS

Let $\mathrm{Attn_m}$ denote the attention matrix at transformer layer $m$ for a sequence of length $L_{seq}$.

**Previous Token Head Strength** (PHStrength) measures attention paid by all the tokens to their immediate predecessor. We define:

$$\mathrm{PHStrength_m^{(1)}} = \frac{1}{L_{seq}-1} \sum_{i=1}^{L_{seq}-1} \mathrm{Attn_m}\big(\mathrm{query}_i \to \mathrm{key}_{i-1}\big).$$
$$(2)$$

In the multimodal interleaving setting, where useful dependencies may shift by an additional offset, we also track attention to the token two positions back: $\mathrm{PHStrength_m^{(2)}}$.

**Induction Head Strength** (IndStrength) quantifies how strongly the target token attends to the labels of the context examples of the same class.

$$\mathrm{IndStrength_m} = \frac{1}{|\mathcal{P}|} \sum_{j \in \mathcal{P}} \mathrm{Attn_m}(\mathrm{query_{target}} \to \mathrm{key}_j),$$
$$(3)$$

where $\mathcal{P}$ is the set of positions immediately after context examples of the same class as the target.

**Target Label Association** (TLA) measures total attention the target token paid to all context label positions.

$$\mathrm{TLA_m} = \sum_{j \in \mathcal{Y}} \mathrm{Attn_m} \left( \mathrm{query_{target}} \to \mathrm{key}_j \right), \qquad (4)$$

where $\mathcal{Y}$ is the set of positions of all the labels in the context.

**Context-label Accuracy** (CLA) measures whether the predicted label appeared in the context, indicating reliance on context. Let $\hat{y}$ be the predicted label, $\{y_i\}_{i=0}^N$ the labels of the $N$ context examples, then

$$\mathrm{CLA} = \mathbb{P}\left(\hat{y} \in \{y_i\}_{i=1}^N\right). \qquad (5)$$

Except for Context Label Accuracy, all metrics are computed separately for each transformer layer.

### 4.4.2. ICL DYNAMICS: FROM FORMATION TO REFINEMENT

Our analysis reveals a distinct two-stage evolution of the ICL circuit. Unimodal pretraining installs the foundational

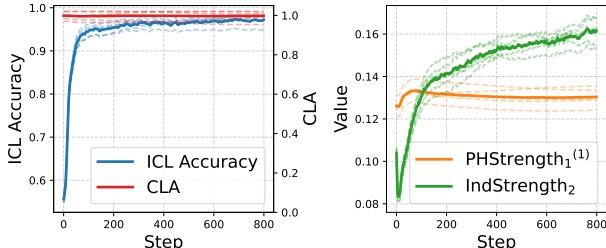

*Figure 9.* Demonstration of the head dynamics in the multimodal setting for five seeds. While CLA and previous token heads remain stable from the unimodal stage, the refinement of the induction head is the primary driver of accuracy gains.

circuit, while multimodal training primarily refines it to accommodate new modalities.

**Correlation analysis reveals a shift in ICL's primary driver.** We compute the Pearson correlation between metrics and ICL accuracy for the runs with varying data statistics for five seeds. The results in Table 1 differ significantly between the two settings. In the unimodal pretraining, $\mathrm{PHStrength_1^{(1)}}$ (previous-token copying) and $\mathrm{IndStrength_2}$ (label matching) show the strongest correlation with ICL accuracy. CLA (predicting the label from the context) is also a strong correlate. This confirms that initial ICL development is driven by learning the foundational mechanisms of an induction circuit: learning the "looking back" and "copy" operation.

However, in the multimodal setting, the dynamics shift. We find that $\mathrm{IndStrength_2}$ becomes the strongest correlate of ICL accuracy. Conversely, the correlation for CLA drops to just 0.02. As shown in Figure 9, this is because the model, having already mastered context-reliance during pretraining, maintains a high CLA ceiling throughout the multimodal stage. Crucially, $\mathrm{PHStrength_1^{(1)}}$ remains a strong correlate, while $\mathrm{PHStrength_1^{(2)}}$ shows negligible correlation. This confirms that the model does not alter its fundamental copying mechanism to span two tokens (skipping the M1 token to look at M2 directly); instead, it reuses the same offset-one circuit established during pretraining. This shift suggests that the model reuses the foundational behaviors learned in pretraining and the new bottleneck for performance is applying this circuit to M2. The model must learn to match the M2 query feature to the context examples, and this "matching" skill is precisely what $\mathrm{IndStrength_2}$ quantifies.

To assess whether these mechanisms are sufficient to explain performance, we split all the runs into training /validation sets and train a random-forest regressor to predict ICL accuracy from the progress measurements. The results in Table 2, demonstrate that the core ICL circuit is highly predictive in both settings. Full analysis see App. A.3.5.

*Table 1.* Pearson correlations ($\rho$) between top progress measurements ($\rho \geq 0.5$) and ICL accuracy. In the unimodal setting, learning foundational mechanisms like previous-token copying and context reliance is dominant. In the multimodal setting, the focus shifts to refining the label-matching induction head.

*(a)* Unimodal Setting

| Metric | $\rho$ |
|---|---|
| $\text{PHStrength}_1^{(1)}$ | 0.72 |
| CLA | 0.65 |
| $\text{IndStrength}_2$ | 0.61 |
| $\text{TLA}_1$ | 0.59 |

*(b)* Multimodal Setting

| Metric | $\rho$ |
|---|---|
| $\text{IndStrength}_2$ | 0.70 |
| $\text{PHStrength}_1^{(1)}$ | 0.58 |
| $\text{TLA}_2$ | 0.56 |
| CLA | 0.02 |

*Table 2.* A random forest regressor can predict final ICL accuracy ($R^2$) from our progress measurements in both settings.

| Feature Subset | Unimodal | Multimodal |
|---|---|---|
| $\text{PHStrength}_1^{(1)}$, $\text{IndStrength}_2$ | $0.91 \pm 0.02$ | $0.90 \pm 0.01$ |
| $\text{PHStrength}_2^{(1)}$, $\text{TLA}_1$, $\text{IndStrength}_2$ | $0.96 \pm 0.02$ | $0.96 \pm 0.06$ |
| All metrics | $0.97 \pm 0.01$ | $0.98 \pm 0.01$ |

### 4.4.3. VALIDATING THE MECHANISM

**Validating Circuit Necessity.** While progress measurements correlate strongly with ICL accuracy, correlation does not imply causation. To validate the mechanistic necessity of these multimodal ICL circuits, we move beyond the correlation analysis and perform targeted causal interventions at inference time. Concretely, we identify the previous token head and induction head (head with max $\text{PHStrength}_1^{(1)}$ and $\text{IndStrength}_2$) and knock them out by zeroing the corresponding attention. The results in Table 3 confirm these circuits are causal drivers of performance.

**Cross-Modal Integration.** To confirm that this circuit refinement represents genuine cross-modal integration rather than a degenerate reliance on the primary modality, we conduct a modality zeroing ablation at inference time. Zeroing M2 reduces ICL accuracy from $> 96\%$ to $33.6\%$, proving induction heads depend on M2 features for discrimination. However, zeroing M1 causes a catastrophic drop to $6.3\%$. These results (App. A.3.6) confirm that M1 provides the structural scaffold; the model cannot perform ICL from M2 alone because the circuit logic remains anchored in the primary modality's embedding space.

> **Takeaway.** Unimodal pretraining constructs the specific ICL circuits, while multimodal training refines them.

## 5. Validation on Production MLLMs

Our controlled testbed makes three sharp predictions about production-scale multimodal models: **(P1) Decoder scale**: unlike the unimodal regime, scaling should consistently improve multimodal ICL (Sec. 4.2); **(P2) Mechanism**: the

*Table 3.* Effect of Circuit Ablation on multimodal ICL.

| Configuration | ICL Accuracy ($\pm\sigma$) |
|---|---|
| Baseline | $0.970 \pm 0.025$ |
| Knocking out Previous Token Head | $0.199 \pm 0.005$ |
| Knocking out Induction Head | $0.062 \pm 0.003$ |

previous-token (PH) and induction (IH) heads identified in our synthetic setting should also exist in LLMs and be inherited by MLLMs. These heads causally drive ICL, and are refined during multimodal training (Sec. 4.4). We verify each prediction below. Architectural details, the head-identification protocol, the LoRA fine-tuning configuration, and full results are deferred to App. A.4.

**(P1) Multimodal ICL improves consistently with decoder scale.** We compare two model families across six VL-ICL subtasks (Zong et al., 2024). Within each family, the larger decoder consistently outperforms the smaller one: Qwen2.5-VL (Team, 2025) gains $+2.3\%$ average accuracy from 3B to 7B, and IDEFICS (Laurençon et al., 2023) gains $+10.5\%$ from 9B to 80B (per-task numbers in Table 10).

**(P2a) Induction circuits are inherited from the LLM backbone.** We rank PH and IH heads independently on Qwen2.5-VL-3B-Instruct and on its text-only backbone Qwen2.5-3B-Instruct, using the Open-MI subtask with $n_{\text{shot}}=2$. The rankings agree substantially: 4 of the MLLM's top-5 PH heads appear in the LLM's top-10, and 7 of its top-10 PH and IH heads appear in the LLM's top-20 (see Table 11). The circuits driving multimodal ICL are largely *inherited* rather than *constructed during* multimodal training.

**(P2b) The identified heads are causally responsible for multimodal ICL.** To distinguish correlation from cause we ablate the identified heads at inference by knocking them out. On 50 held-out Open-MI queries (Table 12), knocking out the top-5 PH heads drops ICL accuracy from $0.74$ to $0.65$; the top-5 IH heads drop it to $0.58$; the combined knockout reaches $0.56$, approaching the random baseline of $0.50$. The IH heads dominate the causal effect, mirroring the synthetic-setting finding that the label-matching induction head is the locus of multimodal ICL.

**(P2c) Multimodal training refines rather than rebuilds the circuit.** We fine-tune Qwen2.5-VL-3B-Instruct on Open-MI and track, every 25 steps, the PHStrength of the top-5 PH heads, the IndStrength of the top-5 IH heads, ICL accuracy, and $CLA$ (the probability the prediction is *any* in-context label). Fig. 18 reproduces the synthetic Stage 2 dynamics: PHStrength is essentially flat across training, IndStrength rises in lockstep with ICL accuracy, and CLA is pinned at the $1.0$ ceiling throughout. The model always

copies from context and fine-tuning only changes *which* in-context label is selected, by sharpening the inherited induction circuit.

## 6. Related Work

**Unimodal ICL.** First observed as an emergent LLM ability (Brown et al., 2020), ICL has since been studied in NLP and synthetic settings (Min et al., 2022; Akyürek et al., 2022; Wei et al., 2023; Von Oswald et al., 2023; Singh et al., 2023; Dong et al., 2024), with work probing the roles of training data and demonstrations (Chan et al., 2022; Shin et al., 2022; Yadlowsky et al., 2023; An et al., 2023; Liu et al., 2024). Mechanistic studies have focused on induction circuits, first discovered by Elhage et al. (2021) and further investigated across various settings (Olsson et al., 2022; Wang et al., 2023), alongside theoretical interpretations (Xie et al., 2021; Singh et al., 2024). Reddy (2024) explored ICL emergence through both induction circuits and data distributional properties using minimal attention-only transformers, which particularly motivated our work. While these mechanistic analyses typically use simplified architectures that diverge from modern LLMs in key components, we further investigate more realistic structures and extend the analysis to the multimodal setting.

**Multimodal ICL.** Existing work on multimodal ICL has primarily focused on understanding or improving the behavior of MLLMs. Studies have documented multimodal ICL capabilities in models pretrained on interleaved image–text corpora (Alayrac et al., 2022; Abdin et al., 2024; Hui et al., 2024; Zong et al., 2024). Others have worked on strengthening this behavior (Zhao et al., 2023; Doveh et al., 2024; Yu et al., 2024; Jiang et al., 2025; Jia et al., 2025; Chen et al., 2025b; Huang et al., 2024). Diagnostic work has revealed that apparent multimodal ICL in MLLMs is often largely text-driven (Chen et al., 2025a; Baldassini et al., 2024). Our work differs fundamentally in scope and objective: rather than analyzing MLLMs or using a simplified model as a proxy to explain MLLM behavior, we study multimodal ICL as a phenomenon in transformers in its own right, contributing to the broader understanding of transformer learning rather than to MLLM interpretability specifically.

## 7. Conclusion

We uncover a learning asymmetry where the primary modality installs the core ICL circuit, enabling multimodal ICL to emerge with low data complexity in the secondary modality. This explains why scaling consistently improves multimodal ICL: the added capacity is used to map the secondary modality to the pre-existing ICL circuit, unlike the unimodal tendency where capacity shifts toward memorization. Mechanistically, both settings rely on induction heads, though RoPE universally suppresses this capability.

## 8. Limitation

Our mechanistic conclusions are established in small, controlled transformers, which is precisely what enables clean causal attribution; correspondingly, transfer to production-scale MLLMs should be interpreted as a qualitative bridge rather than a fully verified one. Likewise, while the GMM testbed likely understates the difficulty of real-world cross-modal alignment, its value is that it isolates distributional factors that are heavily confounded in natural corpora. Finally, our progress measures are best viewed as informative operational probes of circuit formation, rather than exhaustive descriptions of all computations in larger architectures.

## Acknowledgments

This work was partially funded by the ERC (853489 - DEXIM) and the Alfried Krupp von Bohlen und Halbach Foundation, which we thank for their generous support. This work was also supported by Hi! PARIS and ANR/France 2030 program (ANR-23-IACL-0005). We are also grateful for partial support from the Pioneer Centre for AI, DNRF grant number P1. The authors gratefully acknowledge the scientific support and resources of the AI service infrastructure LRZ AI Systems provided by the Leibniz Supercomputing Centre (LRZ) of the Bavarian Academy of Sciences and Humanities (BAdW), funded by Bayerisches Staatsministerium für Wissenschaft und Kunst (StMWK).

## Impact Statement

This work contributes a mechanistic dissection of multimodal in-context learning and an open, low-compute synthetic testbed for isolating how data statistics and architectural choices shape transformer learning. By mapping the high-level capability of cross-modal ICL onto specific, quantifiable attention mechanisms (the refinement of induction heads), we improve the transparency of multimodal large language models and provide researchers with a tractable substrate for studying learning phenomena that are typically obscured by the confounds of large-scale training. We anticipate the testbed will be reused beyond our experiments to investigate circuit-level robustness, the interaction of positional encodings with induction circuits, and architectural designs that favour compositional reasoning over memorisation. We see no specific societal risks or harms beyond those generally associated with transformer-based foundation models.

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

# A. Appendix

## A.1. Preliminaries

In the unimodal setting, inputs consist of $N$ labeled tuples followed by a query (Figure 10a). Figure 10b summarizes the synthetic data generation and distributional controls, and Figure 10c details the IWL/ICL evaluation protocol.

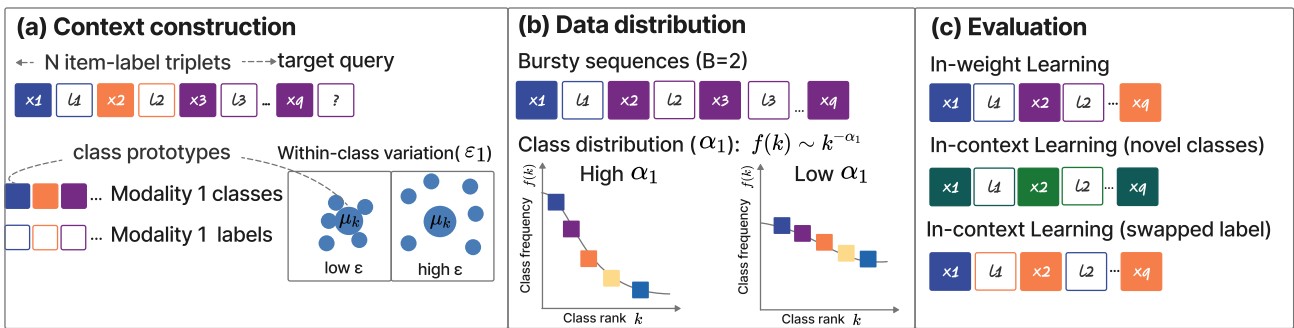

Figure 10. Overview of the preliminaries in the unimodal setting. **(a)** The context consists of $N$ item-label pairs $(x_i, l_i)$ followed by the target query. $K$ classes are assigned to $L$ labels. **(b)** The distributional properties for the synthetic data. Class instances are obtained by controlling within-class variation $\varepsilon$. Class frequencies follow a Zipfian distribution with exponent $\alpha$, and burstiness $B$ determines repeated class occurrences in context. **(c)** Evaluation distinguishes between in-weight learning, where test items belong to seen classes while not in the context, and in-context learning (ICL), where test classes are in the context but novel. A swapped-label condition further isolates ICL by permuting label assignments during evaluation.

**Evaluation metrics.** We separate IWL from ICL to avoid conflating memorization with contextual reasoning, following Reddy (2024); Chan et al. (2022). Each test episode is a sequence of $N$ labeled exemplars (context) followed by a query; unless stated, we match training hyperparameters (e.g., $N$, burstiness $B$, Zipf exponent $\alpha$, within-class variation $\varepsilon$). **IWL:** sample both context items and the query i.i.d. from the *training* class distribution with the same priors; the query's class does not appear in the context examples. **ICL–Novel (primary test):** use novel classes unseen during training, provide labeled exemplars for those classes in the context, and draw the query from the same novel-class set; this measures the ability to infer class–label mappings from context alone. **ICL–Swap (label-swap control):** use training classes but apply a random permutation to context labels at test time and evaluate against the permuted mapping; this invalidates memorized mappings and isolates reliance on context. We report accuracy for IWL, ICL (average of ICL–Novel and ICL–Swap).

## A.2. Extended results and analysis in unimodal experiments

### A.2.1. UNIMODAL DATA DISTRIBUTIONAL PROPERTIES

As shown in Figure 11, key trends are reproduced. In addition, we notice that while higher within-class noise $\varepsilon$ again promotes ICL, we additionally find that it slows convergence, as the model requires more data to form robust associations.

### A.2.2. IMPACT OF POSITIONAL ENCODINGS, CONTEXT LENGTH, AND DATA COMPLEXITY

To further investigate the mechanistic impact of positional encodings (PE) discussed in Sec. 3, we conducted a broader analysis comparing **Absolute PE (APE)**, **RoPE**, **ALiBi**, and a **Hybrid (APE + RoPE)** model. We swept two key variables: the context length (N, the number of item-label pairs) and the data complexity (proxied by $K \cdot \sqrt{B}$). The results are shown in Figure 12. First, for a fixed data complexity, ICL accuracy degrades for *all* PE types as the context length (N) increases. This suggests that longer contexts make it more difficult to find the matching exemplar, increasing the difficulty for the ICL circuit (as supported by the decreasing induction circuit strength in Figure 12). Secondly, We observe that ALiBi and RoPE perform similarly, and as argued in Sec. 3, both appear to weaken the formation of the offset-based induction circuits compared to APE. The Hybrid (APE + RoPE) model's performance lies in between as expected. Third, while APE and Hybrid PE demonstrate stronger ICL performance across most of the tested regimes, it is crucial to note the effect of data complexity. As seen in the high-complexity setting (Figure 12, right), the performance gap narrows significantly. At high complexity and short context lengths (e.g., N=8), it is possible for all four PE types to achieve near-perfect ICL accuracy.

These findings show that the design of RoPE and ALiBi provides a weaker inductive bias for the simple, offset-based

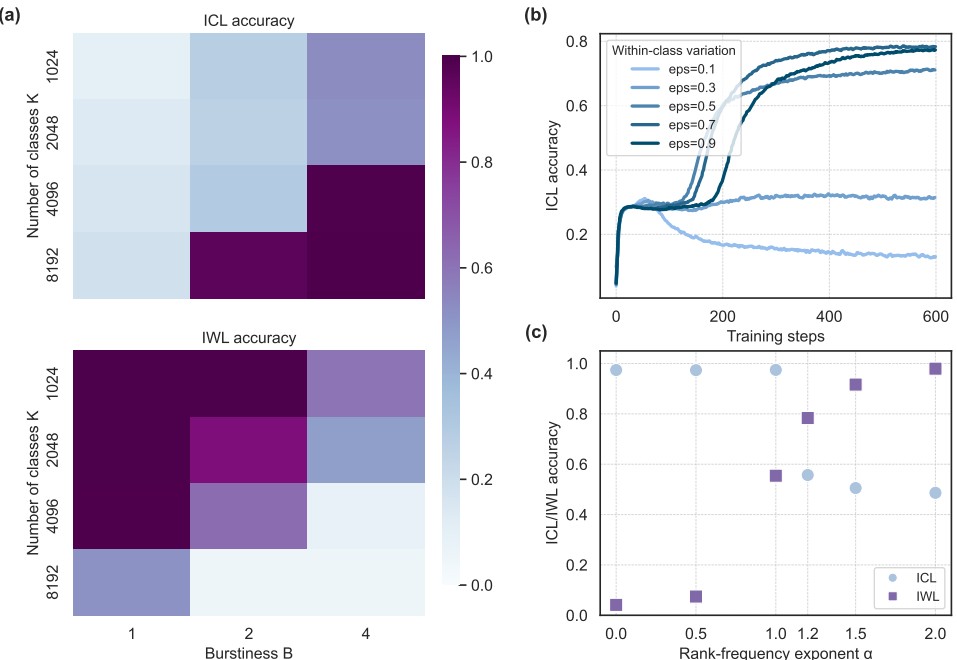

*Figure 11.* The data distributional findings transfer from the GMM setting. *(a) Number of classes and burstiness.* Larger number of classes $K$ and burstiness $B$ promotes ICL while decreasing IWL. *(b) Within-class variation.* Increasing $\varepsilon$ promotes ICL. However, it also slows down the emergence of ICL. *(c) Class-distribution skew.* Increasing $\alpha$ improves ICL. When $\alpha = 1$, a balance is achieved between ICL and IWL.

induction circuits. This effect is most pronounced in low-data-complexity regimes, and *sufficiently high data complexity can compensate for this weaker bias. Importantly, they do not prevent ICL.*

*Table 4.* Pearson correlations ($\rho$) between progress measurements and ICL accuracy. We consider $\rho > 0.5$ strong and $\rho > 0.7$ very strong.

*(a) Unimodal setting*

| Rank | PM | $\rho$ |
|---|---|---|
| 1 | $\text{PHStrength}_1$ | 0.72 |
| 2 | CLA | 0.65 |
| 3 | $\text{IndStrength}_2$ | 0.61 |
| 4 | $\text{TLA}_1$ | 0.59 |
| 5 | $\text{TLA}_2$ | 0.11 |
| 6 | $\text{IndStrength}_1$ | 0.06 |
| 7 | $\text{PHStrength}_2$ | $-0.10$ |

*(b) Multimodal setting*

| Rank | Metric | $\rho$ |
|---|---|---|
| 1 | $\text{IndStrength}_2$ | 0.70 |
| 2 | $\text{PHStrength}_1^{(1)}$ | 0.58 |
| 3 | $\text{TLA}_2$ | 0.56 |
| 4 | $\text{TLA}_1$ | 0.51 |
| 5 | $\text{PHStrength}_1^{(2)}$ | 0.48 |
| 6 | $\text{PHStrength}_2^{(2)}$ | 0.47 |
| 7 | $\text{IndStrength}_1$ | 0.10 |
| 8 | CLA | 0.02 |
| 9 | $\text{PHStrength}_1^{(1)}$ | $-0.02$ |

**Pearson Correlation.** The results in the Table 4a align with the standard induction-circuit picture: $\text{PHStrength}_1$ correlates most with ICL accuracy (previous-token copying), $\text{IndStrength}_2$ ranks highly (label matching), and CLA is also strong, reflecting prediction from context labels. CLA ranks second, indicating that high ICL models tend to predict labels seen in the context. TLA in the first layer shows moderate correlation with ICL accuracy. This suggests that, in addition to the previous token function, the first attention head also contributes to the model learning to predict the labels existing in the context. This is further supported by a Pearson correlation of 0.6 between $\text{TLA}_1$ and CLA, indicating that attention to all context labels in early layers may be a key mechanism behind the model's ability to select labels from the context.

**Random forest prediction.** The results in the Table 5 show that the combination of $\text{PHStrength}_1$ and $\text{IndStrength}_2$, which correspond to the two key components of the induction head circuit, achieves an $R^2$ score of 0.909. This result

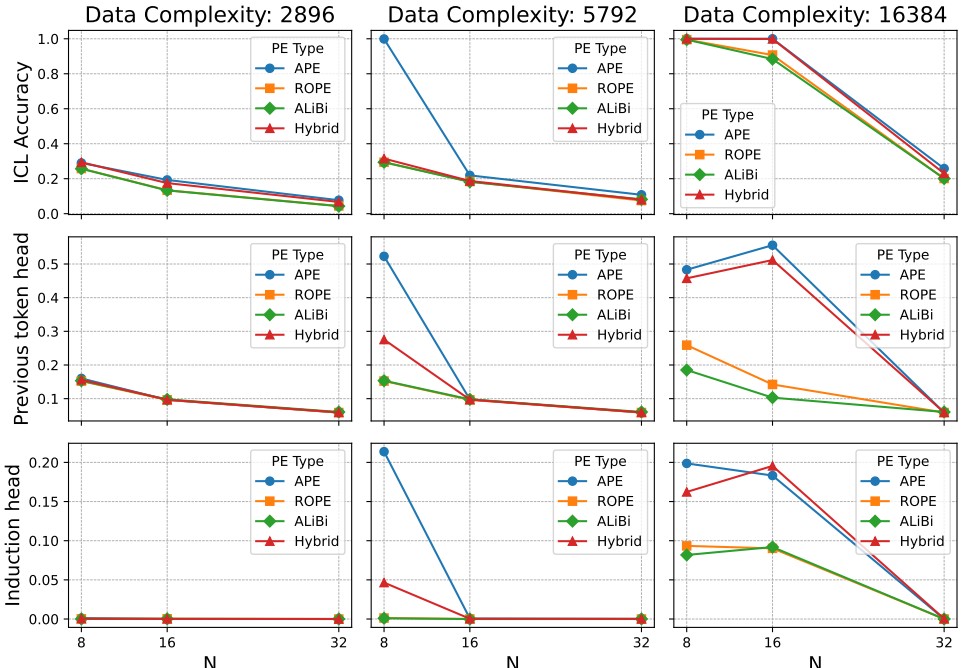

*Figure 12.* Comparison of the impact of different positional encodings across data complexity and context length on ICL accuracy and the strength of previous token head and induction head. Three key patterns are visible: **(1)** ICL accuracy degrades for all PE types as context length (N) increases. **(2)** ALiBi and RoPE cluster together, while APE shows the highest accuracy, with the Hybrid PE performing in between. **(3)** The performance gap between encodings narrows significantly in the high-complexity regime, where all PEs can achieve near-perfect accuracy at short context lengths.

indicates that these two mechanistic metrics alone capture the majority of the variance in ICL accuracy across models trained with varying data distributions and architectures. Adding $\text{TLA}_1$ as a third feature further improves performance to $R^2 = 0.960$. This suggests that the attention to all label positions provides complementary information to the mechanistic induction circuit, likely supporting the model's broader ability to locate context-relevant labels. This interpretation is reinforced by the earlier finding that $\text{TLA}_1$ correlates moderately with both ICL accuracy and CLA. Using the full set of progress measurements, the regressor achieves an $R^2$ of $0.974 \pm 0.014$, indicating that ICL performance is not only correlated with, but also highly predictable from, internal attention behaviors. These results suggest that attention patterns such as previous-token copying, label-focused attention, and context alignment are not incidental byproducts of learning, but form a reliable, quantifiable foundation for in-context reasoning.

*Table 5.* Random-forest performance ($R^2$) for predicting unimodal ICL accuracy from subsets of progress measurements.

| Feature subset | $R^2$ (mean $\pm$ std) |
|---|---|
| $\text{PHStrength}_1$, $\text{IndStrength}_2$ | $0.91 \pm 0.02$ |
| $\text{PHStrength}_1$, $\text{TLA}_1$, $\text{IndStrength}_2$ | $0.96 \pm 0.02$ |
| All metrics | $0.97 \pm 0.01$ |

### A.3. Extended results in multimodal experiments

#### A.3.1. RoPE CONTINUES TO RAISE THE DATA COMPLEXITY THRESHOLD FOR MULTIMODAL ICL.

Consistent with our unimodal results, Rope continues to raise the data complexity threshold for multimodal ICL to emerge, as shown in the Figure 13.

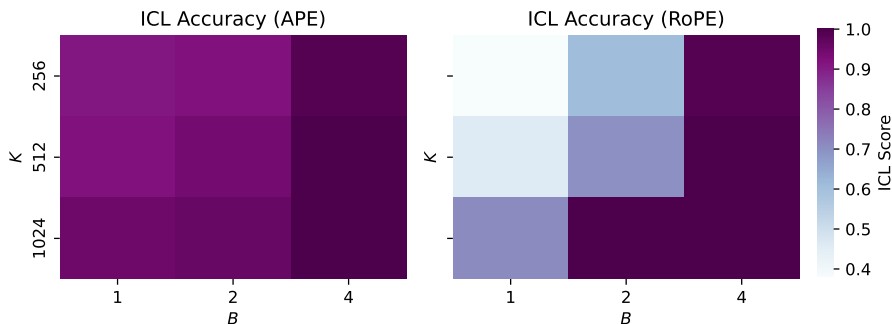

*Figure 13.* In the multimodal setting, RoPE yields lower ICL accuracy than APE across data regimes. Consistent with unimodal setting, RoPE increases the data complexity threshold for strong ICL.

### A.3.2. IMPACT OF FEATURE DIMENSIONALITY ON CROSS-MODAL ALIGNMENT

To investigate the relationship between feature dimensionality and ICL performance, we measured the alignment between the projected M2 features and M1 class prototypes using Centered Kernel Alignment (CKA) and Euclidean Distance ($L_2$). The results are shown in the Table 6. As the feature dimension $D_2$ increases from 32 to 512, we observe a consistent degradation in alignment metrics: CKA values drop and $L_2$ distances increase (Row 1 vs. Row 4). While introducing the encoder consistently improves alignment compared to the "Projector Only" baseline (e.g., at $D_2 = 512$, CKA improves from 0.07 to 0.10 and $L_2$ decreases from 2.15 to 1.95), it does not fully restore the metrics to the levels observed at lower dimensions ($D_2 = 32$). This indicates that while the encoder mitigates misalignment, higher dimensionality introduces structural discrepancies that persist even with the encoder.

*Table 6.* Higher M2 feature dimensions($D_2$) degrade alignment (CKA $\downarrow$, $L_2$ $\uparrow$). Here we use the feature after the projector for both with and without encoder settings. We fix $D_1 = 64$.

| $D_2$ | CKA w/o Enc | CKA w/ Enc | L2 w/o Enc | L2 w/ Enc |
|---|---|---|---|---|
| 32 | 0.16 | 0.17 | 0.95 | 0.91 |
| 128 | 0.12 | 0.14 | 1.47 | 1.37 |
| 256 | 0.09 | 0.11 | 1.82 | 1.43 |
| 512 | 0.07 | 0.11 | 2.15 | 1.45 |

### A.3.3. DATA DISTRIBUTIONAL FINDINGS TRANSFER TO REAL IMAGE DATA.

We extend our experiments on Omniglot dataset, which has 1.6k handwritten characters. We pre-train the decoder on GMM data and construct the multimodal input by varying the number of classes, the burstiness, within-class variance (adding noise) ,and the class distribution. The results in Figure 14 show that findings from the synthetic setting transfer.

**Mini-ImageNet validation.** To verify that the data-complexity asymmetry survives the move to natural-image content with much richer visual structure than Omniglot, we repeat the Stage 2 experiment with Mini-ImageNet (Russakovsky et al., 2015) as the M2. We pretrain a ViT-Small encoder (patch size 7, image size 84, 3 channels) on the 64 Mini-ImageNet train classes, freeze it, and train a 2-layer projector with a frozen pretrained 2-layer decoder. We fix $K_1=8192$ (synthetic Gaussian M1, identical to the main experiments) and sweep $K_2 \in \{16, 32, 64\}$ and $B \in \{1, 2, 4\}$. Figure 15 reproduces the two core trends from Fig. 4: larger $K_2$ and $B$ both improve ICL accuracy and suppress IWL; and a $K_2$ as small as 16 already yields non-trivial multimodal ICL ($\sim 0.58$ at $B{=}4$), confirming that the asymmetry is not an artefact of either synthetic or low-complexity image data.

### A.3.4. HOW DOES THE PRETRAINING DATASET FOR THE ENCODER IMPACT DOWNSTREAM MULTIMODAL ICL?

We pretrain a family of encoders on Omniglot with controlled difficulty (class noise, sample count, label skew, regularization), yielding a range of validation accuracies. Each encoder is attached to the same projector and pretrained decoder, then fine-tuned under identical regimes. Encoder validation accuracy strongly predicts downstream multimodal ICL, whereas

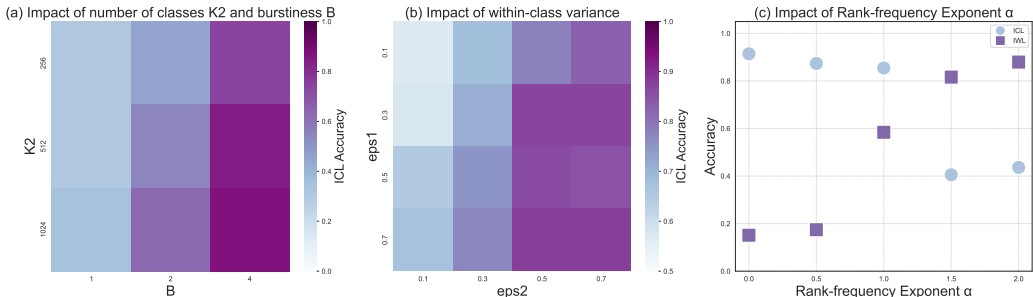

*Figure 14.* The data distributional findings transfer from the GMM setting to the **Omniglot** dataset. *(a)* Larger number of classes $K2$ and burstiness $B$ promotes ICL. The symmetry also transfers. When $K1 = 8192$, a smaller $K_2 = 256$ is enough for the model to learn a very good ICL. *(b)* Within-class variation. Increasing $\varepsilon_1$ and $\varepsilon_2$ promotes ICL. *(c)* Class-distribution skew. Increasing $\alpha$ improves ICL. When $\alpha_1 = \alpha_2 = 1$, a balance is achieved between ICL and IWL.

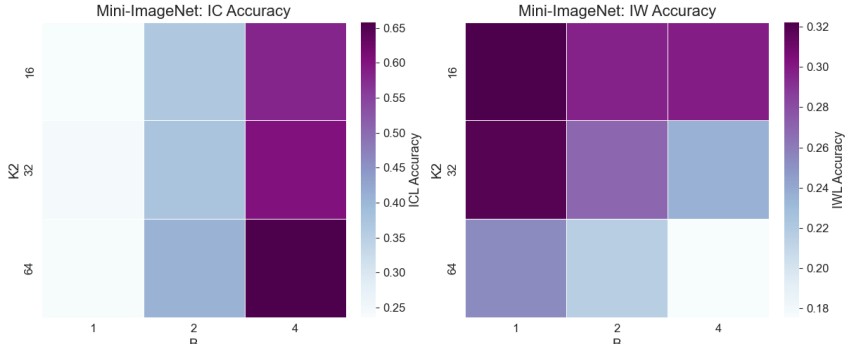

*Figure 15.* **Mini-ImageNet** validation of the data-complexity asymmetry. $K_1$=8192 on synthetic Gaussian M1; we vary $K_2$ (Mini-ImageNet train classes) and burstiness $B$. The asymmetry continues to hold on natural images: small $K_2$ suffices once the M1 decoder is pretrained.

encoder size and the specific pretraining statistics do not show a clear independent effect (Figs. 16, 17).

### A.3.5. PROGRESS MEASUREMENT ANALYSIS

**Pearson Correlation.** As shown in the Table 4b, $\mathrm{PHStrength}_1^{(1)}$ shows weaker correlation with ICL compared with $\mathrm{IndStrength}_2$. During the unimodal pretraining stage, the model already learns a strong previous-token head in the first layer which is retained in the multimodal stage, as shown in the Figure 9. It indicates that in the multimodal training stage, the dominant learning objective is to match the new modality to the correct class, making the continued development of the induction head the most sensitive indicator of progress. Similarly, $CLA$ exhibits a very low correlation with ICL accuracy because the model has learned in the first stage to predict the label from the context. In the multimodal training, $CLA$ maintains at a very high value as shown in the Figure 9. $\mathrm{TLA}_1$ continues to play a role as in the unimodal setting and interestingly, $\mathrm{TLA}_2$ shows a moderately strong correlation with ICL accuracy ($r = 0.54$), but we find that it correlates even more strongly with $\mathrm{IndStrength}_2$ ($r = 0.84$). This suggests that the predictive power of $\mathrm{TLA}_2$ on ICL may be largely mediated through its alignment with $\mathrm{IndStrength}_2$, which is the dominant indicator of ICL performance in this stage ($r = 0.68$). In this view, $\mathrm{TLA}_2$ does not independently drive ICL, but rather co-develops with induction behavior. As the model learns to retrieve the correct label via contextual matching (induction), it also increases its attention to label positions more generally. This leads to a shared rise in both metrics, resulting in high intercorrelation and a secondary correlation with ICL. Thus, $\mathrm{TLA}_2$'s relevance to ICL may be best understood as a supporting mechanism that reflects the formation of stronger induction behavior in the second layer.

**Random forest prediction.** The results in Table 7 confirm that the core induction mechanism remains the primary driver: using just $\mathrm{PHStrength}_1^{(1)}$ and $\mathrm{IndStrength}_2$ achieves a strong predictive performance with $R^2 = 0.9$. Adding $\mathrm{TLA}_1$ improves the $R^2$ to 0.96. In contrast, adding $\mathrm{TLA}_2$ results in a smaller gain ($R^2 = 0.92$), which supports our interpretation that its contribution is largely redundant with $\mathrm{IndStrength}_2$. Using all available metrics yields $R^2 = 0.98$, showing that

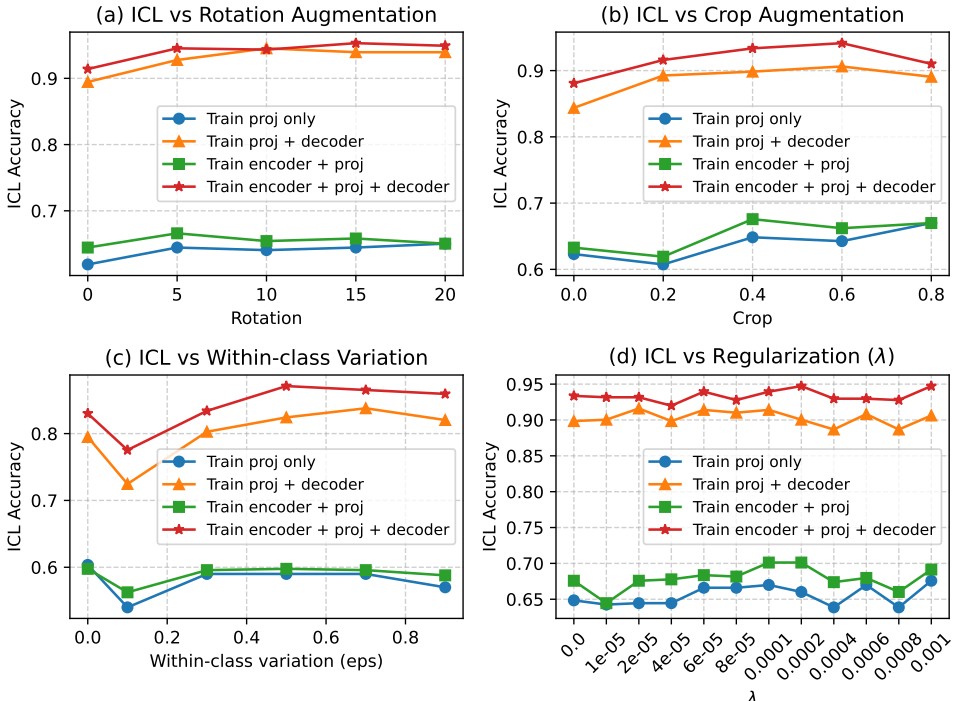

*Figure 16.* The correlation between statistical properties and regularization during encoder pretraining with the downstream ICL does not show a clear pattern across different training regimes.

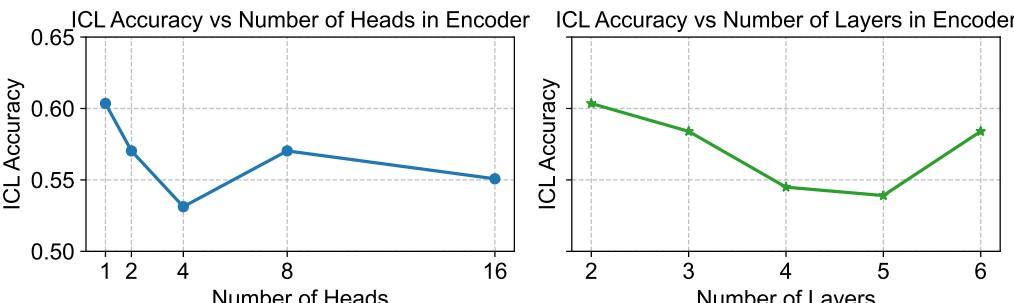

*Figure 17.* The encoder size shows no clear correlation with multimodal ICL accuracy.

multimodal ICL accuracy can be predicted with high fidelity from a small set of interpretable attention behaviors.

*Table 7.* RF performance ($R^2$) using subsets of the progress measurements in the multimodal setting.

| Feature subset | $R^2$ (mean $\pm$ std) |
|---|---|
| $\text{PHStrength}_1^{(1)}, \text{IndStrength}_2$ | $0.90 \pm 0.01$ |
| $\text{PHStrength}_1^{(1)}, \text{IndStrength}_2, \text{TLA}_1$ | $0.96 \pm 0.06$ |
| $\text{PHStrength}_1^{(1)}, \text{IndStrength}_2, \text{TLA}_2$ | $0.92 \pm 0.01$ |
| All metrics | $0.98 \pm 0.01$ |

### A.3.6. CROSS-MODAL INTERACTION ANALYSIS VIA MODALITY ZEROING

To directly test whether the model performs genuine cross-modal reasoning or simply relies on a degenerate M1-only strategy, we conduct an ablation study where we selectively remove one modality at test time by replacing its tokens with

zero vectors.

We take a model trained on multimodal sequences $x_1, x'_1, \ell_1, x_2, x'_2, \ell_2, \ldots, x_q, x'_q$ and evaluate it under three conditions:

- **Full (M1 + M2)**: standard evaluation with both modalities present

- **M1 only (M1 + zeros)**: replace all M2 tokens $x'_i$ with zero vectors

- **M2 only (zeros + M2)**: replace all M1 tokens $x_i$ with zero vectors

Table 8 shows that zeroing either modality substantially degrades ICL accuracy, confirming that the model relies on both modalities during inference. Zeroing M2 reduces accuracy to 0.336 ($-65.2\%$), showing that removing the secondary modality's information substantially harms performance. Zeroing M1 is even more severe, reducing accuracy to 0.063 ($-93.5\%$), which indicates that while the induction circuit was initially installed by M1 during pretraining, M2 alone cannot recover ICL without the primary modality's contextual structure. These results, combined with our progress measurement analysis in Sec. 4.4.2, demonstrate that multimodal ICL is not a degenerate M1-only process. The model genuinely integrates information from both modalities through the shared self-attention mechanism.

*Table 8.* Modality ablation results. ICL accuracy drops dramatically when either modality is removed, confirming genuine cross-modal integration.

| Input Type | M1 + M2 | M1 + zeros | zeros + M2 |
|---|---|---|---|
| ICL Accuracy | 0.967 | 0.336 | 0.063 |

### A.3.7. EARLY-FUSION JOINT TRAINING

In the main paper, we train the decoder on M1 (unimodal pretraining) before introducing M2 via a projector (late fusion). To understand how the observed modality asymmetry depends on this training schedule, we conducted additional experiments with **early-fusion joint training**, where both modalities are trained together from scratch.

We initialize a transformer from scratch and train it jointly on both M1 and M2. Since M1 and M2 have different dimensionalities ($D_1 \neq D_2$), we use a simple projector to map M2 into M1's embedding space. The input sequence is $x_1, x'_1, \ell_1, x_2, x'_2, \ell_2, \ldots, x_q, x'_q$, where $x_i$ are M1 tokens and $x'_i$ are projected M2 tokens. We systematically vary the number of classes in each modality ($K_1, K_2$) and the burstiness $B$, then measure ICL and IWL accuracies.

Table 9 reports the corresponding ICL and IWL accuracies. The results reveal a reversal of the asymmetry observed in late-fusion training. In late fusion, where M1 pretraining installs the induction circuit before M2 is introduced, varying $K_1$ has a stronger effect on ICL than varying $K_2$. In early fusion, the opposite is true: varying $K_2$ shows a much stronger effect on ICL performance across all burstiness levels. This reversal is explained by the positional structure of the sequence. In our early-fusion setup, each label $\ell_i$ is positioned immediately after the M2 token $x'_i$ rather than the M1 token $x_i$. Consequently, the simplest previous-token pattern the model can learn is to attend from $\ell_i$ to $x'_i$, and the simplest induction pattern for the query token is to locate the matching $x'_i$ (in M2) and move one step forward to retrieve the label. This architectural design naturally encourages the induction circuit to anchor on M2. The primary-modality asymmetry observed in Sec. 4.1 is not a fixed property of the modalities themselves but emerges from the combination of pretraining schedule (which modality first learns the circuit) and sequence geometry (which tokens are positionally adjacent to labels). In late-fusion architectures with unimodal pretraining, the text modality becomes primary because it installs the induction circuit during language pretraining. Early-fusion joint training can shift this role to the vision modality if the sequence structure favors it.

### A.4. Detailed setup and full results for Qwen2.5-VL validation

This appendix expands on the headline results in Sec. 5. Each block below states the protocol, reports the full numerical result, and interprets it in the light of the corresponding synthetic claim.

**Decoder scaling on the VL-ICL benchmark.** We evaluate two MLLM families on the six VL-ICL subtasks: Qwen2.5-VL-3B vs. Qwen2.5-VL-7B (Team, 2025), and IDEFICS-9B vs. IDEFICS2-80B (Laurençon et al., 2023).Both families improve with decoder size on the benchmark (Table 10); the per-family average accuracies improve by $+2.33\%$ (Qwen, 3B→7B) and $+10.52\%$ (IDEFICS, 9B→80B). This matches the synthetic prediction of Sec. 4.2.

*Table 9.* **Early fusion performance analysis.** A comparison of In-Context Learning (ICL) and In-Weights Learning (IWL) accuracy across varying number of classes $(K_1, K_2)$ and burstiness $(B)$.

| | $B = 1$ | | | | $B = 2$ | | | | $B = 4$ | | | |
|---|---|---|---|---|---|---|---|---|---|---|---|---|
| $K_2\backslash K_1$ | 2048 | 4096 | 8192 | 16384 | 2048 | 4096 | 8192 | 16384 | 2048 | 4096 | 8192 | 16384 |
| *In-Context Learning (ICL) Accuracy* | | | | | | | | | | | | |
| 2048 | 0.25 | 0.25 | 0.27 | 0.25 | 0.37 | 0.39 | 0.39 | 0.38 | 0.59 | 0.57 | 0.56 | 0.56 |
| 4096 | 0.28 | 0.26 | 0.25 | 0.26 | 0.37 | 0.39 | 0.37 | 0.35 | 0.57 | 0.58 | 0.59 | 0.56 |
| 8192 | 0.27 | 0.27 | 0.28 | 0.29 | 0.62 | 0.37 | 0.37 | 0.35 | 0.88 | 0.95 | 0.88 | 0.96 |
| 16384 | 0.26 | 0.27 | 0.29 | 0.31 | 0.52 | 0.65 | 0.50 | 0.57 | 0.89 | 0.87 | 0.58 | 0.88 |
| *In-Weights Learning (IWL) Accuracy* | | | | | | | | | | | | |
| 2048 | 1.00 | 0.99 | 0.99 | 0.99 | 0.97 | 0.98 | 0.98 | 0.97 | 0.67 | 0.68 | 0.65 | 0.65 |
| 4096 | 0.98 | 0.97 | 0.97 | 0.98 | 0.70 | 0.70 | 0.73 | 0.73 | 0.46 | 0.41 | 0.38 | 0.43 |
| 8192 | 0.24 | 0.24 | 0.23 | 0.24 | 0.09 | 0.12 | 0.15 | 0.13 | 0.09 | 0.08 | 0.08 | 0.10 |
| 16384 | 0.10 | 0.10 | 0.10 | 0.09 | 0.09 | 0.08 | 0.08 | 0.09 | 0.07 | 0.07 | 0.09 | 0.08 |

*Table 10.* **Per-task VL-ICL accuracy (%) for Qwen2.5-VL-3B/7B and IDEFICS-9B/80B.** Both families improve with decoder scale, supporting the prediction that scaling consistently benefits multimodal ICL.

| Task | Qwen-3B | Qwen-7B | IDEFICS-9B | IDEFICS-80B |
|---|---|---|---|---|
| CLEVR | 5.50 | 7.50 | 30.33 | 32.43 |
| Matching-MI | 54.25 | 58.75 | 0.00 | 28.30 |
| Open-MI | 58.75 | 62.75 | 59.17 | 61.50 |
| Operator Induction | 8.35 | 8.50 | 14.44 | 21.67 |
| Operator Induction (Interleaved) | 10.00 | 11.35 | 15.00 | 36.67 |
| TextOCR | 46.50 | 48.50 | 28.00 | 29.50 |
| **Average** | **30.56** | **32.89** $(+2.33 \uparrow)$ | **24.49** | **35.01** $(+10.52 \uparrow)$ |

**Head-analysis architecture and metric definitions.** The head-level analyses use Qwen2.5-VL-3B-Instruct together with its text-only backbone Qwen2.5-3B-Instruct as comparator. Both models have $L=36$ layers and $H=16$ query heads. For every layer-head pair $(\ell, h)$ we compute PHStrength and IndStrength. The IndStrength averages the last-token attention to answer-token positions(located by sub-sequence search on both the bare and space-prefixed BPE encodings of the label). Ranks are taken on per-example means and aggregated across the evaluation batch. MLLM head identification uses 16 Open-MI image-ICL examples; LLM head identification uses 50 text-only examples built by substituting each support image with its `real_name` description while keeping the chat-template structure identical.

**Inheritance of induction circuits.** We rank PH and IH heads independently on the LLM and the MLLM with the metrics above. Table 11 lists the top-5 PH and IH heads in each model (bold = overlap with the counterpart's top-5). Across the broader top-$K$, the rankings agree substantially: $4/5$ of the MLLM's top-5 PH heads lie in the LLM's top-10 PH ranking; $7/10$ of the top-10 PH heads lie in the LLM's top-20. The results show that the MLLM inherits the existing language-side circuit and uses it, which is a direct mechanistic counterpart to the synthetic-setting finding that multimodal training reuses the pretrained PH head verbatim and refines the existing IH head (Sec. 4.4). The full top-20 rankings are released as `top_heads.json`/`top_heads_llm.json`.

**Causal knockout.** For each head $(\ell, h)$ to be ablated we register a forward pre-hook on `layers[ℓ].self_attn.o_proj` that zeros the input slice corresponding to head $h$ (positions $h \cdot d_{\text{head}} : (h+1) \cdot d_{\text{head}}$), removing that head's contribution to the residual stream while leaving the rest of the model unchanged. We evaluate on 50 held-out Open-MI queries. Qwen2.5-VL-3B reaches ICL accuracy 0.74. Knocking out the top-5 PH heads drops accuracy to 0.65; the top-5 IH heads drop it to 0.58; the combined PH+IH knockout reaches 0.56, approaching the binary-task random baseline of 0.50 (see Table 12). The IH heads are causally responsible for multimodal-ICL: removing them collapses performance to near random even though the rest of the model is untouched and a fully grounded visual representation reaches the decoder. The head-level account from Sec. 4.4 is the right level of description even at production scale.

**Fine-tuning dynamics.** We LoRA (Hu et al., 2022) fine-tune Qwen2.5-VL-3B-Instruct on Open-MI with rank $r=16$, scale

*Table 11.* **Top-5 PH and IH heads identified independently in Qwen2.5-VL-3B (MLLM) and Qwen2.5-3B (LLM).** Bold = overlap with the counterpart model's top-5.

| Rank | MLLM PH (L,H) | LLM PH (L,H) |
|------|---------------|--------------|
| 1 | **(19,11)** | **(1, 1)** |
| 2 | (19, 1) | **(19,11)** |
| 3 | **(31, 9)** | (28, 2) |
| 4 | **(1, 1)** | (23,15) |
| 5 | **(2,13)** | **(31, 9)** |

| **Induction heads** | | |
|------|---------------|--------------|
| Rank | MLLM IH (L,H) | LLM IH (L,H) |
| 1 | (31,15) | (20, 1) |
| 2 | (31, 8) | **(25, 5)** |
| 3 | (31,12) | (26,15) |
| 4 | **(30, 3)** | **(30, 3)** |
| 5 | **(25, 5)** | (20, 4) |

*Table 12.* Head knockout on Qwen2.5-VL-3B (VL-ICL Open-MI). Top-5 PH/IH heads identified in Sec. 5 are ablated.

| Condition | ICL acc |
|-----------|---------|
| Qwen2.5-VL-3B | 0.74 |
| Knock out top-5 PH heads | 0.65 |
| Knock out top-5 IH heads | 0.58 |
| Knock out top-5 PH + IH heads | 0.56 |
| Random baseline | 0.50 |

$\alpha$=32, dropout 0.05, target modules $\{$q, k, v, o$\}$_proj, and bias frozen. Optimiser: AdamW with lr=$2 \times 10^{-5}$, batch size 1. The training loss is masked to the answer + EOS tokens (prompt tokens excluded). We run 500 steps with evaluation every 25 steps. Train / eval split: 100 / 16 Open-MI examples. At each evaluation we record the PHStrength of the top-5 PH heads, the IndStrength of the top-5 IH heads, ICL accuracy, and Context-Label Accuracy (CLA: the probability the prediction is *any* in-context label). Fig. 18 shows that ICL accuracy rises from a non-trivial starting value (the model already does some Open-MI ICL out of the box) to a substantially higher plateau over 500 steps. The average PHStrength of the tracked heads stays essentially flat, while the average IndStrength rises in lockstep with ICL accuracy. CLA is pinned at the 1.0 ceiling throughout. The results show that multimodal fine-tuning does not invent a new copying mechanism; CLA at the ceiling shows the model always copies from context, and PH stability shows the offset-1 copying head is unchanged. What fine-tuning does is sharpen the inherited IH heads so that the right in-context label is selected. This is identical to the synthetic Stage 2 dynamics reported in Sec. 4.4 and constitutes a strong end-to-end test of our circuit-refinement account at production scale.

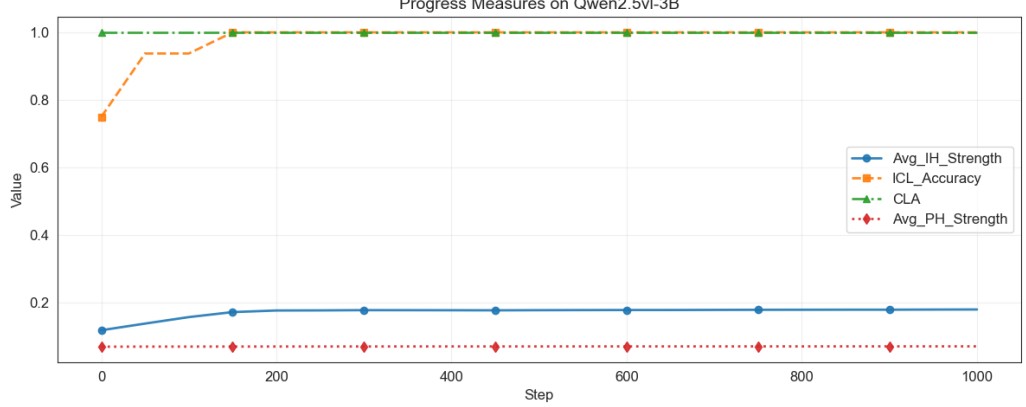

*Figure 18.* The finetuning dynamic of Qwen2.5-VL-3B-Instruct on Open-MI.

