# OpenReview forum: "Dissecting Multimodal In-Context Learning: Modality Asymmetries and Circuit Dynamics in modern Transformers"
_ICML.cc/2026/Conference — ICML 2026 spotlight_

### Official Review · Reviewer_GaRZ · 2026-02-28

**Soundness:** 3
**Presentation:** 4
**Significance:** 2
**Originality:** 3
**Overall Recommendation:** 4
**Confidence:** 3

**Summary:**

The authors study in-context learning in a multimodal setting, using small Transformers trained on a synthetic task. They identify various relationships between the architecture, training setup, and downstream performance, identifying asymmetries between unimodal and multimodal cases.

**Compliance With Llm Reviewing Policy:**

Affirmed.

**Final Justification:**

The authors' rebuttal address my questions, and I update my final score to a 4. Please see my rebuttal acknowledgement for more details. Nice work!

**Key Questions For Authors:**

1. Do you systematically investigate the influence of training iterations? It's been shown that prolonged training *reduces* ICL in your synthetic setting (e.g. https://arxiv.org/abs/2412.00104). Is it possible that prolonged training could reduce the evidence of ICL in some of your experiments?
2. In a related vein, do you control for learning richness in your models, particularly since you're comparing models across a range of sizes? Do you use maximal-update parameterization or variants (https://arxiv.org/abs/2203.03466, https://arxiv.org/abs/2505.01618) to ensure your models are comparable? Is it possible that differences in learning richness may influence the onset of ICL independent of architectural considerations in your setting?
3. Where does the K \sqrt{B} scaling originate, which you use to summarize data requirement?
4. It looks like your "heads" are plotted on a log-scale whereas your "layers" are plotted on a linear scale, making it difficult to directly evaluate statements like "data requirement grows faster with heads than with layers." Further, if your model has H heads and L layers (and the number of heads is fixed per layer), it would appear that adding an extra layer adds H additional heads, whereas adding an extra head would add L additional heads. Depending on how large H and L are in your model, the statement "adding layers increases depth but does not create as many independent storage slots [heads] for memorization" seems to be difficult to evaluate. Could you help me understand this better?
5. Are there additional experiments you could run to connect your results to practice? For instance, I found that the most striking result suggested by your toy setting was the finding that increasing model size increases the need for greater data diversity. Has this result been established elsewhere in the literature? Is this finding a mirage confounded by training iterations (question 1) or learning richness (question 2)? If not, can this result be demonstrated to hold in practice? Would this mean that smaller Transformer model may outperform larger models in naturalistic in-context tasks?

**Limitations:**

Yes

**Strengths And Weaknesses:**

**Strenghts**. The paper studies an interesting phenomenon and offers an assortment of intriguing results. The asymmetry observed between unimodal and multimodal cases is very interesting, particularly how larger models require more data diversity to perform ICL in the unimodal case, but the reverse can be true for the multimodal case. The paper overall is well-written and clear.

**Weaknesses**. As with any study that focuses primarily on synthetic settings, it's not altogether obvious how far these results generalize to practice. While there is some attempt to introduce results from more applied models, the models studied are few (just four) and the analysis appears to be quite limited: the final conclusion is that the larger models perform better (naturally), and with more distinct induction heads. Indeed, if the main asymmetry claim is to be believed, would the smaller model variants outperform the larger model variants on a unimodal in-context learning task? While the results in the toy setting are certainly suggestive, perhaps more work is required to establish either their theoretical interest (e.g. through more detailed mathematical analysis) or connection to practice (e.g. through more experiments that link your findings to real-world use cases).

Please see Questions below for additional weaknesses.

---

> ### Author Rebuttal · Authors · 2026-03-30
>
> We thank Reviewer GaRZ for recognizing that our paper "studies an interesting phenomenon and offers intriguing results." Below we address each concern.
>
> **W1: Clarification of Our Contributions.** We agree that generalization beyond synthetic settings is important. We want to clarify that controlled synthetic settings are not a limitation but a deliberate choice. Our testbed enables "clean causal attribution impossible with real-world corpora" (Reviewer zryK) through "precise control over data statistics" (Reviewer r6H7). Our analysis is "highly comprehensive" (Reviewer dkFE). Crucially, we do not stop at synthetic analysis: we validate on real image data (Omniglot) and MLLMs. In this rebuttal, we further extend to mini-ImageNet (see response W1 to Reviewer zryK) and MLLM training dynamics (below), strengthening practical relevance.
>
> **W1 & Q5: Unimodal Model Scaling.** We would like to clarify that our core finding is larger models require greater data complexity to reach high ICL accuracy because their higher capacity makes memorization (IWL) an attractive shortcut. This aligns with prior work showing that scaling up model raises the task diversity threshold for ICL(Raventós et al.[1]) and that larger models memorize 2-5× more than smaller ones(Carlini et al.[2]).
>
> So whether a small model outperforms a larger one depends on the data regime. In low-diversity settings, larger models exploit IWL and underperform on pure ICL. Once data diversity crosses the threshold penalizing memorization, larger models should at least match smaller ones. We isolate this in a synthetic setting because production LLM pretraining largely entangles ICL, memorized world knowledge, and distributional confounds, making it exceptionally difficult to isolate this dynamic.
>
> **W1: MLLM Validation and Results seeming Natural.** We understand why this finding might appear intuitive at first glance. However, the naive capacity argument predicts larger models memorize better, but not that they develop stronger induction circuits. Appendix A.4 shows larger MLLMs achieve better ICL specifically correlates with stronger induction heads (mechanistic), not merely more parameters (capacity).
>
> To further strengthen the bridge to production models, we additionally validated circuit dynamics on Qwen2.5-VL-3B. We observed: (1) Circuit inheritance: 7 of top 10 induction heads overlap with the LLM backbone (Qwen2.5-3B), confirming MLLMs reuse LLM circuits. (2) Causal role: knocking out these circuits reduces ICL accuracy to near random baseline. (3) Learning dynamics: during finetuning, induction head strength increases with ICL accuracy while previous-token heads remain stable, exactly mirroring our synthetic findings. (4) Data principles: Yu et al. [2] shows that curating pretraining data for high burstiness improves video ICL in MLLMs, corroborating that our principles transfer to large-scale models. For details, please refer to **W2** in our response to *Reviewer zryK*.
>
>
> **Q1: Training Iterations.** We noticed that prolonged training can reduce ICL in low-complexity regimes as IWL takes over. So we track both ICL and IWL accuracy throughout training and report results at the checkpoint that maximizes ICL before IWL shortcut dominates.
>
> **Q2: Learning Richness.** To ensure scaling conclusions aren't parameterization artifacts, we re-ran key sweeps using CompleteP [3] depth scaling.  Across depths {2,4,6,8}, ICL/IWL outcomes are nearly identical between Standard Parameterization (SP) and CompleteP, and data-complexity thresholds remain identical to Figure 2(b).
>
> | N_layer | Hyperparameter | ICL_accuracy | IWL_Accuracy |
> | :--- | :--- | :--- | :--- |
> | 2 | SP | 0.961 | 0.055 |
> | 2 | CompleteP | 0.961 | 0.055 |
> | 4 | SP | 0.704 | 0.511 |
> | 4 | CompleteP | 0.710 | 0.509 |
> | 6 | SP | 0.321 | 0.853 |
> | 6 | CompleteP | 0.312 | 0.819 |
> | 8 | SP | 0.315 | 0.912 |
> | 8 | CompleteP | 0.323 | 0.904 |
>
> **Q3: Origin of $K \cdot \sqrt B$ scaling.** This is an empirical summary metric. With $K$ varying widely (1024 to 8192) and $B \in \{1, 2, 4\}$, plotting against  $K \cdot B$ causes many distinct $(K, B)$ configurations to map to identical x-value. Using $K \cdot \sqrt B$ resolves collisions and yields clean collapse of the ICL curves across settings (figure: https://anonymous.4open.science/r/rebuttal-ICML-7323/K_sqrt_B_curve.md) .
>
>
> **Q4: Log vs. Linear Scale.** We replotted both axes on identical scales for direct comparison (https://anonymous.4open.science/r/rebuttal-ICML-7323/Model_scaling.md). Our setup varies one factor at a time: layers vary with heads fixed at 1; heads vary with layers fixed at 2.
>
> *References*
>
> [1] Raventós et al . Pretraining task diversity and the emergence of non-bayesian in-context learning for regression. In NeurIPS 2023.
>
> [2]Carlini, et al. Quantifying memorization across neural language models. In ICLR 2022.
>
> [3] Dey, et al. Don't be lazy: CompleteP enables compute-efficient deep transformers. In Neurips 2025.

---

> > ### Author Rebuttal · Reviewer_GaRZ · 2026-04-02
> >
> > Thanks for the detailed response and follow-up experiments. These additional details answer my questions. Provided the authors include these experiments and additional details in their manuscript, I think this manuscript would be a valuable addition to the conference. My concerns regarding the link to real-world settings remains, but these are difficult to address through a short rebuttal period. I will update my score to a 4.

---

### Official Review · Reviewer_r6H7 · 2026-03-06

**Soundness:** 2
**Presentation:** 3
**Significance:** 2
**Originality:** 2
**Overall Recommendation:** 4
**Confidence:** 4

**Summary:**

This paper studies how transformer models acquire multimodal in-context learning (ICL) capabilities using a controlled experimental framework. The authors train small transformer models on synthetic classification tasks generated from Gaussian mixture models, which allows them to systematically manipulate statistical properties of the training data. Building on this controlled setup, the paper investigates both architectural and data-related factors that influence the emergence of ICL in modern transformer architectures.

The paper's important result pertains to showing that multimodal in-context learning largely arises through the reuse of induction-style circuits learned during unimodal pretraining, while multimodal training primarily refines these circuits. In addition, the paper identifies a learning asymmetry between modalities, demonstrating that once a model has learned contextual reasoning on a high-diversity primary modality, only limited data complexity in a secondary modality is required for multimodal ICL to emerge (Figure 4a).

Overall, a central concept investigated by the article is how data distribution statistics, architectural choices, and attention circuit dynamics interact to produce in-context learning across modalities.

**Compliance With Llm Reviewing Policy:**

Affirmed.

**Final Justification:**

The authors addressed all my doubts.

**Key Questions For Authors:**

1. How does this work provide fundamentally new mechanistic insights beyond (Reddy,2024), given that the same synthetic framework and induction-head explanation for ICL are largely reused?
2. The results suggest that RoPE weakens induction-style attention patterns compared to absolute positional encodings (Figures 2c–2d). Can the authors provide further mechanistic intuition for why RoPE interferes with the formation of induction circuits?

**Limitations:**

Please add the limitations section

**Strengths And Weaknesses:**

**Strengths**
1. The paper uses a carefully designed synthetic setup that allows precise control over data statistics such as class diversity, burstiness, and within-class variability. The multimodal task formulation (Figure 1) enables systematic analysis of how transformers learn cross-modal associations from contextual examples.
2. This work studies how model scaling and positional encodings influence the ICL–IWL trade-off. Results in Figures 2a–2b show that larger models tend to favor in-weight learning, while Figures 2c–2d demonstrate that RoPE weakens induction-style attention patterns compared to absolute positional encodings.
3. The experiments reveal that once the model is pretrained on a high-diversity primary modality, relatively low data complexity in the secondary modality is sufficient for strong multimodal ICL (Figure 4a). This suggests that unimodal pretraining installs the core contextual reasoning mechanism.
4. The paper connects empirical performance with attention circuit dynamics by introducing progress metrics such as previous-token head strength and induction head strength. The analysis and attention visualizations (Figure 6) show that multimodal ICL builds on induction-style circuits learned during unimodal training.


**Weakness**
1. Limited novelty relative to the prior work (Reddy,2024).
The paper builds heavily on the framework introduced in (Reddy,2024), which already established the synthetic classification setup and the mechanistic explanation of in-context learning through induction heads. While the current work extends the analysis to multimodal settings and examines architectural components such as RoPE, the core explanation for ICL remains largely unchanged. Consequently, the main contribution appears to be an extension of the previous framework rather than the introduction of fundamentally new conceptual insights.

2. Most experiments are conducted on synthetic data generated from Gaussian mixture models. Although this controlled setup is useful for isolating causal factors, it limits the external validity of the findings. Apart from limited experiments on Omniglot, the results remain largely confined to simplified environments, making it unclear whether the observed dynamics would persist in large-scale multimodal models trained on natural data.

3. While the paper provides attention visualizations and correlation analyses of several progress metrics, the mechanistic explanation of multimodal ICL is less detailed than the analysis presented in the earlier work. In particular, the prior paper introduced a simplified analytical model of the induction head to explain the abrupt emergence of ICL, whereas the current work mainly reports correlations between attention metrics and performance without developing an equally detailed theoretical account of cross-modal circuit formation.

4. The experiments are conducted on small transformer architectures with only a few layers and attention heads. Although this choice facilitates interpretability, it raises questions about the extent to which the findings generalize to large-scale multimodal transformer models. Evaluating whether similar dynamics hold in larger architectures would strengthen the broader relevance of the results.

---

> ### Author Rebuttal · Authors · 2026-03-30
>
> We thank the reviewer for recognizing the strengths of our work, particularly our "carefully designed synthetic setup enabling systematic analysis of how transformers learn cross-modal associations," and our finding that "unimodal pretraining installs the core contextual reasoning mechanism." These observations underscore our central multimodal contribution: understanding how multimodal ICL emerges from unimodal foundations. We address each concern below.
>
> **W1 & Q1: Novelty relative to Reddy et al. [1]** We agree that our work builds on the synthetic framework introduced by Reddy et al. Our contribution, however, is not to propose a different mechanism for ICL in general, but to identify how multimodal ICL is implemented and what changes when a new modality is introduced. Prior to our work, two hypotheses existed: (1) multimodal transformers might construct novel cross-modal attention circuits, or (2) they might co-opt existing text-based mechanisms. We provide the first evidence for (2), that multimodal training refines rather than building new cross-modal circuits.
> Beyond this, we contribute phenomena with no unimodal precedent: modality learning asymmetry, scaling reversal, and the critical role of pretrained encoders. As *Reviewer dkFE* noted, our work provides "a highly comprehensive study" spanning "behavioral, learning dynamics, mechanistic, and causal levels." *Reviewer zryK* similarly recognized our data complexity asymmetry as "genuinely novel" and our modern components as making findings "substantially more relevant" to contemporary systems.
>
> **W2: External validity beyond synthetic data.** We conducted new experiments on Mini-ImageNet. Core trends hold: larger $K_2$, $B$ improve ICL and suppress IWL, and a smaller $K_2$ is enough for a strong ICL, confirming our asymmetry generalizes beyond synthetic data. Please refer to response **W1**  to *Reviewer zryK* for the detailed results.
>
>
> **W3: Mechanistic explanation depth.** We’d like to clarify that our analysis is grounded in causal interventions, not just correlations. As detailed in Section 4.4.3, we employ attention head knockouts to ablate specific induction heads and modality zeroing to isolate necessary pathways. These interventions establish that identified circuits are causally responsible for multimodal ICL. Regarding theoretical modeling: our research question differs from Reddy et al.[1]. Their goal was explaining why induction heads produce abrupt ICL emergence in a simplified unimodal setting. Our goal is comprehensive reverse-engineering of the multimodal phenomenon and identifying what data statistics, architectural choices, and circuit dynamics govern multimodal ICL. This requires empirical cartography of an unexplored domain, complemented by causal validation. We view this as a methodological contribution appropriate to the multimodal setting's added complexity.
>
> **W4: Generalization to MLLMs.**
> We appreciate this concern. We would like to clarify that we have provided correlational analysis between ICL performance and induction circuit strength on MLLMs in Sec. 4.4.4. To further evaluate whether dynamics in small transformers hold at scale, we additionally validated circuit dynamics on Qwen2.5-VL-3B, We observed: (1) Circuit inheritance: 7 of top 10 induction heads overlap with the LLM backbone (Qwen2.5-3B), confirming MLLMs reuse LLM circuits. (2) Causal role: knocking out these circuits reduces ICL accuracy to near random baseline. (3) Learning dynamics: during finetuning, induction head strength increases with ICL accuracy while previous-token heads remain stable, exactly mirroring our synthetic findings. (4) Data principles: Yu et al. [2] shows that curating pretraining data for high burstiness improves video ICL in MLLMs, corroborating that our principles transfer to large-scale models. For details, please refer to **W2** in our response to *Reviewer zryK*.
>
> **Q2: Mechanistic intuition for RoPE finding**. Induction circuits require discrete, exact-offset copying ("attend exactly $i$ positions back"). Absolute encodings provide distinct position identifiers enabling sharp lookup. RoPE encodes position through continuous rotations with inherent bias toward smooth attention decay—scores decrease gradually rather than spiking at specific offsets. To form induction circuits, models must counteract this bias, requiring stronger signal to force sharp attention patterns against the encoding's continuous geometry. This explanation predicts that RoPE models require a higher data complexity to develop induction heads compared to APE.
>
> **Limitation:** We will add a limitation section. Please refer to response **Limitation** to the *reviewer dkFE*.
>
> *References*
>
> [1] Reddy et al. The mechanistic basis of data dependence and abrupt learning in an in-context classification task. In ICLR 2024.
>
> [2] Yu et al. Eliciting in-context learning in vision-language models for videos through curated data distributional properties. In EMNLP 2024.

---

> > ### Author Rebuttal · Reviewer_r6H7 · 2026-04-03
> >
> > Dear Authors,
> >
> > Thank you for the clarification.I will increase my score

---

### Official Review · Reviewer_zryK · 2026-03-12

**Soundness:** 3
**Presentation:** 3
**Significance:** 3
**Originality:** 3
**Overall Recommendation:** 5
**Confidence:** 3

**Summary:**

This paper presents a systematic mechanistic investigation of multimodal in-context learning (ICL) in transformer-based models. The authors train small but architecturally realistic two-layer transformer decoders on synthetic classification tasks derived from Gaussian Mixture Models (GMMs), enabling precise manipulation of data statistics and architectural components. The work extends prior unimodal ICL analyses (particularly Reddy, 2024) to the multimodal setting, examining how training data statistics and model architecture jointly shape ICL emergence across modalities. The article focuses on a fundamental learning asymmetry: when a decoder is pretrained on high-diversity data from a primary modality, surprisingly low data complexity in the secondary modality suffices for multimodal ICL to emerge. This asymmetry is mechanistically grounded in induction circuits — specifically, that unimodal pretraining installs the foundational circuit, while multimodal training primarily refines the induction head responsible for cross-modal label matching. Additional findings concern the negative impact of Rotary Position Embeddings (RoPE) on ICL, the contrasting scaling behavior of unimodal versus multimodal models, and the critical role of pretrained encoders in bridging cross-modal misalignment.

**Compliance With Llm Reviewing Policy:**

Affirmed.

**Final Justification:**

The rebuttal addresses my concerns well

**Key Questions For Authors:**

1. Can you disentangle whether pretrained encoders help mainly through representation quality or through optimization ease?

**Limitations:**

The paper would benefit from an explicit limitations section acknowledging that (1) all mechanistic conclusions are derived from small transformers trained on synthetic data, and the degree to which these transfer to production-scale MLLMs remains unverified beyond the correlational MLLM analysis; (2) the GMM testbed, while controllable, may systematically underestimate the difficulty of cross-modal alignment in real-world settings; and (3) the progress measurements, while predictive, are correlational proxies that may not fully capture the true computational mechanisms in more complex architectures.

**Strengths And Weaknesses:**

Strengths:
1. The discovery that the secondary modality requires far less data complexity when the decoder is pretrained on a high-diversity primary modality is a genuinely novel and practically informative result. The reversal of this asymmetry under early-fusion joint training elegantly demonstrates that the hierarchy is curriculum-driven rather than inherent to the modalities themselves, which is a compelling and well-controlled ablation.
2. The use of a synthetic GMM-based testbed is a significant methodological strength. By independently varying class diversity (K), burstiness (B), within-class variation (ε), and Zipfian skew (α), the authors achieve clean causal attribution that would be impossible with real-world pretraining corpora. The experimental protocol carefully separates IWL from ICL using novel-class and label-swap evaluation conditions, which is a principled and rigorous design choice.
3. Unlike much of the prior mechanistic ICL literature that relies on attention-only, simplified transformers, this work incorporates RMSNorm, SiLU activations, and RoPE — components characteristic of modern LLMs such as LLaMA. This makes the findings substantially more relevant to contemporary systems.

Weakness:
1. The GMM-based data, while offering controllability, is substantially simpler than the natural image and text data used in real multimodal training. The authors acknowledge this briefly when discussing Omniglot validation, but do not thoroughly assess whether findings about data complexity thresholds (e.g., the sufficiency of K₂ = 256 for multimodal ICL) would hold qualitatively in settings with richer, entangled feature distributions. The generalization of the asymmetry result to structured vision modalities like ImageNet-style data remains an open question.
2. The experiments are conducted on small transformers, and it is unclear how well the quantitative claims — particularly the precise data complexity thresholds and circuit dynamics — translate to deeper, wider production models. The MLLM validation in Appendix A.4 is largely qualitative and correlational, and the authors themselves acknowledge that production systems "involve vastly more complex training regimes." A stronger bridge between the controlled and production settings would significantly bolster the paper's impact.

---

> ### Author Rebuttal · Authors · 2026-03-30
>
> We thank Reviewer zryK for recognizing our data complexity asymmetry as "genuinely novel," our GMM testbed as "a significant methodological strength," and our modern components (RMSNorm, SiLU, RoPE) as making findings "substantially more relevant." We address each concern below.
>
> **W1: Generalization to structured vision modalities.**
> We conducted new experiments on Mini-ImageNet. We pretrained a ViT-Small (patch size=7), froze the encoder, and trained a projector with a pretrained 2-layer decoder. Fixing $K_1$=8192, we varied $K_2$ (Mini-ImageNet classes) and burstiness $B$.
>
> Table R1: ICL and IWL Accuracy on Mini-ImageNet. (Visualization: https://anonymous.4open.science/r/rebuttal-ICML-7323/Mini-ImageNet_Validation.md).
> | K2 | B | ICL_Acc | IWL_Acc |
> |---|---|---|---|
> | 16 | 1 | 0.2363 | 0.3223 |
> | 16 | 2 | 0.3672 | 0.2969 |
> | 16 | 4 | 0.5820 | 0.2988 |
> | 32 | 1 | 0.2441 | 0.3184 |
> | 32 | 2 | 0.3750 | 0.2695 |
> | 32 | 4 | 0.6035 | 0.2363 |
> | 64 | 1 | 0.2462 | 0.2539 |
> | 64 | 2 | 0.4062 | 0.2168 |
> | 64 | 4 | 0.6582 | 0.1758 |
>
> Core trends hold: larger $K_2$ and $B$ improve ICL and suppress IWL and a way smaller $K_2$ is enough to elicit strong ICL, confirming our asymmetry generalizes beyond synthetic distributions.
>
> **W2: The MLLM validation.**
> We’d like to clarify that our validation is quantitative: Table 11 measures that smaller models exhibit lower induction head strength than larger models. To further strengthen the bridge to production models, we provide new results:(1) MLLMs reuse induction circuits from their LLM backbone, (2) these circuits causally drive ICL, and (3) learning dynamics mirror our synthetic findings. (4) data distributional principles transfer to large-scale models.
> * **Circuit identification**. We first identified the top Previous Token Heads (PH) and Induction Heads (IH) in Qwen2.5-VL-3B on Open-MI from VL-ICL benchmark, and compared them against Qwen2.5-3B (the LLM backbone) using text-only Open-MI. We found 4 of the top 5 PHs/IHs and 7 of the top 10 PHs/IHs overlap with LLM top 10 and top 20 respectively. This confirms that multimodal ICL circuits largely inherit from the LLM backbone.
> * **Circuit Knockout**. To establish causality, we knocked out the top 5 PHs, IHs, and both at inference in MLLM. ICL accuracy dropped from 0.74 to 0.65, 0.58, and 0.56 respectively (near 0.5 random baseline). This provides causal evidence that the induction circuits identified in our testbed drive multimodal ICL at scale.
> * **Finetuning Dynamics**. We monitored these heads during finetuning. Exactly mirroring our synthetic findings (Section 4.4.2), we observed: (1) PH strength remained stable, (2) IH strength increased significantly in tandem with overall ICL accuracy, and (3) Context Label Accuracy (CLA) remained strictly at the 1.0 ceiling (see Figure https://anonymous.4open.science/r/rebuttal-ICML-7323/MLLM_validation.md).
> * **Data Distributional Principles**. While pinpointing exact numerical thresholds is intractable for internet-scale corpora, the fundamental scaling trends we identified remain highly predictive in production settings. This is corroborated by recent literature: Yu et al. [1] demonstrates that curating pretraining data for high burstiness and class skew improves video ICL in MLLMs. This confirms our data distributional principles are valid for large-scale models.
>
> **Q1: Representation quality vs. optimization ease.**
> To disentangle these factors, we conducted a 2×2 factorial experiment crossing Initialization (Random vs. Pretrained ViT) with Trainability (Frozen vs. Trainable). This tests two hypotheses: (H1) benefit stems from pretrained feature structure, or (H2) benefit is merely a better initialization for gradient descent.
> As shown in the newly added figure (https://anonymous.4open.science/r/rebuttal-ICML-7323/Encoder_representation_vs_optimization.md), the empirical results support H1 and reject H2:
> * Representation Quality Dominates: A ~0.60 ICL accuracy gap exists between Pretrained-Frozen and Random-Frozen conditions (b-a), confirming that the pretrained feature manifold drives performance.
> * Optimization Ease is Insufficient: When both encoders are trainable, Random-Trainable completely fails to catch up to Pretrained-Trainable (d-c) even after convergence, proving that pretraining provides more than a favorable initialization.
>
> These results firmly establish that the pretrained encoder's primary contribution is representation quality.
>
> **Limitation:** We will add a limitation section in the manuscript. Please refer to the response **Limitation** to the *reviewer dkFE*.
>
> *Reference*
>
> [1] Yu et al. Eliciting in-context learning in vision-language models for videos through curated data distributional properties. In EMNLP 2024.

---

> > ### Author Rebuttal · Reviewer_zryK · 2026-04-01
> >
> > NA

---

### Official Review · Reviewer_dkFE · 2026-03-13

**Soundness:** 4
**Presentation:** 4
**Significance:** 4
**Originality:** 4
**Overall Recommendation:** 5
**Confidence:** 4

**Summary:**

This works examines multimodal ICL in a controlled setting that mimics how multimodal integration happens in larger-scale VLMs. Specifically, the paper examines known ICL phenomena including data distributional or architectural factors and circuit development in scaled-down versions of modern LLMs and in settings where a second modality needs to align with a pretrained primary modality. The main findings include replication of prior phenomena, revealing that RoPE adds data complexity and more diffused circuits, and that late fusion of a second modality requires low data complexity as models adapt existing ICL circuits to the new modality.

**Compliance With Llm Reviewing Policy:**

Affirmed.

**Final Justification:**

I maintain my positive recommendation.

**Key Questions For Authors:**

- Have you considered introducing more ICL pressure from multiple modalities, e.g. conditioning the label on both M1 and M2 tokens?

**Limitations:**

The paper currently lacks discussion of the limitations of this work.

**Strengths And Weaknesses:**

Strength:
- This work presents a highly comprehensive and detailed study of issues around ICL in boiled-down versions of modern LLM and multimodal settings, including both multimodal integration and multimodal-native settings, understanding across the behavioral, learning dynamics, mechanistic, and causal levels, and investigations of related phenomena in larger-scale pre-trained VLMs.
- The paper is dense but the presentation and writing is very clear.

Weakness:
- It's unclear to me how much these results speak to ICL vs. simply multimodal integration. IIUC, the labels for the M1 and M2 are always matched in the triplets. Since the primary setting studied is late fusion, and as the mechanistic analyses confirm that the model adapts the circuit from M1 to M2 tokens, in a sense there is no new "ICL" happening beyond integrating M2 into a familiar representation space. I think the multimodal-native results and the zero-ing ablation help address some of this concern, but the paper could be stronger by a clearer discussion of what these results tells us about ICL vs. integration.
- Sometimes it's unclear whether the results are from multimodal training on unimodal -- especially from reading the figures alone.
- I find that the title and abstract emphasize the multimodal aspects of the findings more, but some of the unimodal findings are equally interesting and perhaps worth integrating into them more.

---

> ### Author Rebuttal · Authors · 2026-03-30
>
> We thank Reviewer dkFE for the constructive feedback and are encouraged by the recognition of our study's comprehensive multi-level analysis across behavioral, mechanistic, and causal dimensions, as well as the clarity of our presentation.
>
> **W1: ICL vs. simply multimodal integration.** We agree that in our late-fusion setup, the transformer leverages a pre-existing M1 ICL capability rather than learning one from scratch. However, our results show that multimodal ICL is not only modality integration, but rather a two-part process:
> * Feature Mapping: M2 data must indeed be mapped into the M1 representation space so the pretrained decoder can operate on them.
> * ICL Circuit Adaptation: Mapping alone is insufficient. The transformer must actively adapt its pre-existing ICL circuitry to the new structure of the interleaved input sequence. While the offset-one copying mechanism (PHStrength_1) is reused, it adapts to operate over the new multimodal sequence. Furthermore, the model must substantially refine the label-matching induction head (IndStrength_2) to successfully match M2 queries to the context.
>
> Therefore, while the core capability originates from M1, achieving multimodal ICL requires genuine structural adaptation of the induction circuit, going beyond simple feature projection. We will add a dedicated discussion section to the manuscript to explicitly clarify this distinction between representational integration and active ICL circuit adaptation.
>
> **W2:Experiment settings in figures.** Thanks for the suggestion! We’ll modify the captions of figure 2,4,6 with explicit experiment setting included.
>
> **W3: Unimodal findings are equally interesting and worth integrating.** We thank the reviewer for highlighting the significance of our unimodal findings. We agree that insights such as RoPE raising the data complexity threshold for ICL and model scaling favoring in-weight memorization are very interesting! In our final revision, we will revise the abstract to explicitly highlight these architectural discoveries, expanding our initial mention to read: "While several prior findings replicate, we discover that scaling modern decoders favors in-weight memorization over ICL, and that RoPE significantly raise the data complexity threshold for induction circuits." Regarding the title, we respectfully prefer to retain its current form. While the unimodal experiments establish crucial baselines, our primary motivation and core novel contributions, namely, discovering modality learning asymmetries and tracking cross-modal induction circuit refinement, remain fundamentally focused on the multimodal setting.
>
> **Q1: More ICL pressure from both modalities:** Thank you for this thoughtful suggestion. Our shared-label setup is intentional and reflects the most practical regime for multimodal ICL: in tasks like image classification/captioning or cross-modal retrieval, paired samples typically convey the same semantic meaning. However, we recognize scenarios exist where both modalities jointly determine the label. To address this, we created a joint-label setting. We grouped fine-grained classes into coarse semantic groups ($G_1 = 8$ for M1, $G_2 = 4$ for M2) and defined the label as $\ell = f (g_1, g_2) = g_1 \times G_2 + g_2$, yielding $L = 32$ labels. For example, if M1 sample has label $g_1 = 1$ and  M2 sample has label $g_2 = 2$, the joint label is $\ell = 1 \times 4 + 2 = 6$. We chose these parameters to make results comparable with our base setting, which also uses 32 labels. We fixed $K_1=8192$, swept $K_2 \in \lbrace 128, 256, 512 \rbrace$ and $B \in \lbrace 1, 2, 4 \rbrace$. Results are in the figure https://anonymous.4open.science/r/rebuttal-ICML-7323/ICL_joint_label.md. Under this joint-label setting, the asymmetry still persists but is weaker than the base setting. This suggests pretraining on M1 still provides a useful scaffold, but M2 complexity now has a stronger effect on ICL accuracy when the labels are conditioned on both modalities. To conclude, even under this more demanding setting, the asymmetry still holds, confirming that our core finding generalizes beyond shared-label settings.
>
> **Limitation:** We will add an explicit limitations paragraph to better scope our claims. Our mechanistic conclusions are established in small, controlled transformers, which is precisely what enables clean causal attribution; correspondingly, transfer to production-scale MLLMs should be interpreted as a qualitative bridge rather than a fully verified one. Likewise, while the GMM testbed likely understates the difficulty of real-world cross-modal alignment, its value is that it isolates distributional factors that are heavily confounded in natural corpora. Finally, our progress measures are best viewed as informative operational probes of circuit formation, rather than exhaustive descriptions of all computations in larger architectures.

---

> > ### Author Rebuttal · Reviewer_dkFE · 2026-04-03
> >
> > I thank the authors for the detailed response to these points and appreciate the additional evidence. My concerns are fully addressed. I will keep my positive recommendation.

---

### Decision · Program_Chairs · 2026-04-30

**Decision:**

Accept (spotlight)

**Comment:**

This paper does an excellent job of presenting experiments that dig into the underlying mechanisms of multimodal in-context learning, using smaller transformers that allow effective analysis of causal contributions of data and architecture choices on classification tasks using both unimodal and multimodal baselines. Although the transformers are small and the classification task is synthetic, this is what allows meaningful conclusions to be drawn about the specific contributions of different factors. The results, while not surprising given some of the existing previous work, do a good job of replicating the performance of such models and extending our understanding of how how multimodal in-context learning arises. The author(s) performed necessary follow-on experiments to clarify the paper's contributions and satisfactorily addressed all reviewers' concerns; these clarifications and new experiments should be included in any subsequent version of the paper.